# Size Generalization of Graph Neural Networks on Biological Data: Insights and Practices from the Spectral Perspective

## Abstract

We investigate size-induced distribution shifts in graphs and assess their impact on the ability of graph neural networks (GNNs) to generalize to larger graphs relative to the training data. Existing literature presents conflicting conclusions on GNNs' size generalizability, primarily due to disparities in application domains and underlying assumptions concerning size-induced distribution shifts. Motivated by this, we take a data-driven approach: we focus on real biological datasets and seek to characterize the types of size-induced distribution shifts. Diverging from prior approaches, we adopt a spectral perspective and identify that spectrum differences induced by size are related to differences in subgraph patterns (e.g., average cycle lengths). We further find that common GNNs cannot capture these subgraph patterns, resulting in performance decline when testing on larger graphs. Based on these spectral insights, we introduce and compare three model-agnostic strategies aimed at making GNNs aware of important subgraph patterns to enhance their size generalizability: self-supervision, augmentation, and size-insensitive attention. Our empirical results reveal that all strategies enhance GNNs' size generalizability, with simple size-insensitive attention surprisingly emerging as the most effective method. Notably, this strategy substantially enhances graph classification performance on large test graphs, which are 2-10 times larger than the training graphs, resulting in an improvement in $F_1$ scores by up to 8%.

## 1 Introduction

Graph neural networks (GNNs) (17; 28; 12; 40; 37; 19) have gained widespread popularity in graph classification tasks owing to their outstanding performance. Though most GNNs can process graphs of varying sizes, it remains under-explored whether they can generalize to graphs larger than those encountered during training (size generalizability). Size generalization in GNNs holds significant importance across multiple domains. For instance, in graph algorithmic reasoning (36; 29), GNNs are expected to learn complex algorithms from small examples and generalize that reasoning to larger graphs, as obtaining exact solutions for larger graphs is challenging. In the realm of biology, datasets exhibit a wide range of graph sizes, spanning from small molecules to large compounds. Evaluating whether learned knowledge is influenced by graph size is crucial, as size-dependent information may potentially have a detrimental impact on performance when employing pre-training strategies (13).

Existing literature presents conflicting conclusions on GNNs' size generalizability. On one hand, several studies (21; 35; 26) have provided support for the ability of GNNs to effectively generalize across varying sizes. For instance, a theoretical study (21) established that spectral GNNs exhibit robust transferability between graphs with different sizes and topologies, provided that these graphs discretize the same underlying space in some generic sense. Other works further provided empirical evidence supporting the strong size generalizability of GNNs in the domains of algorithmic task learning (35) and physics simulations (26). On the other hand, several studies (38; 6; 4) have observed performance degradation when a size shift exists between the training and test data. For instance, a recent work (38) showed theoretically and empirically that the difference in degree patterns between small and large graphs contributes to this performance decline. There have also been proposals of novel models (4) and regularization techniques (6) to enhance the size generalizability of GNNs.

These conflicts mainly arise from disparities in application domains and underlying assumptions concerning size-induced distribution shifts.

Motivated by these conflicts, we take a data-driven approach: we focus on real biological datasets and seek to characterize the types of size-induced distribution shifts. This characterization provides valuable insights into the size generalizability of GNNs. Specifically, we adopt a spectral perspective and identify the connections between the spectrum differences induced by varying graph sizes and the differences in subgraph patterns, particularly cycles. We find that breaking cycles in graphs amplifies the spectrum difference between smaller and larger graphs, whereas extending cycle lengths in smaller graphs to align with those in larger graphs reduces this difference. Furthermore, we observe that conventional GNNs struggle to generalize effectively without explicit cycle information, leading to performance degradation on larger graphs. To address this, we propose and compare three model-agnostic strategies aimed at equipping GNNs with cycle information to enhance size generalizability: self-supervision, augmentation, and size-insensitive attention. Our empirical results demonstrate that all strategies enhance GNNs' size generalizability, with simple size-insensitive attention surprisingly emerging as the most effective method. Although prior research has established GNNs' limitations in counting cycles [8], the primary focus of this paper is to delve into how this limitation influences the size generalizability of GNNs.

In sum, our paper makes the following contributions:

- **New Observations.** We characterize the types of distribution shifts caused by various graph sizes in biological networks, offering insights for designing a size-agnostic GNN.
- **Spectral Analysis.** Unlike prior work, we leverage spectral analysis to deepen our understanding of the size generalizability of GNNs.
- **Model Agnostic Strategies.** To make GNNs aware of important size-related subgraph patterns (e.g., average cycle lengths), we propose and compare three model-agnostic strategies that improve size-generalizability of GNNs. We find that simple size-insensitive attention is the most effective strategy among the three.

## 2 NOTATIONS AND PRELIMINARIES

In this section, we begin by introducing the notations and definitions used throughout the paper. Next, we provide an introduction to the fundamentals of GNNs.

### 2.1 NOTATIONS & DEFINITIONS

Let $\mathcal{G}(\mathcal{V}, \mathcal{E})$ be an undirected and unweighted graph with $N$ nodes, where $\mathcal{V}$ denotes the node set, and $\mathcal{E}$ denotes the edge set. The neighborhood of a node $v_i$ is defined as the set of all nodes that connect to $v_i$: $\mathcal{N}_i = \{v_j | (v_j, v_i) \in \mathcal{E}\}$. The graph is represented by its adjacency matrix $\mathbf{A} \in \mathbb{R}^{N \times N}$, and it has a degree matrix $\mathbf{D}$, where the $i$th diagonal element $d_i$ corresponds to the degree of node $v_i$.

**Cycle basis.** An important concept we use to study cycles is *cycle basis* [27]. A cycle basis is defined as the smallest set of cycles where any cycle in the graph can be expressed as a sum of cycles from this basis, similar to the concept of a basis in vector spaces. Here, the summation of cycles is defined as "exclusive or" of the edges. We represent the cycle basis for a graph as $\mathcal{C}$ and refer to the $j$th cycle in this cycle basis as $\mathcal{C}_j$. The cycle basis can be found using the algorithm CACM 491 [25].

### 2.2 GRAPH LEARNING TASK

In this paper, we focus on the graph classification task, where each node $v_i$ is associated with a feature vector $\mathbf{x}_i^{(0)}$, and the feature matrix $\mathbf{X}^{(0)}$ is constructed by arranging the node feature vectors as rows. When using a GNN for the graph classification task, we further denote the node representation matrix at the $l$-th layer as $\mathbf{X}^{(l)}$, and the representation of node $v_i$ as $\mathbf{x}_i^{(l)}$.

**Supervised Graph Classification.** Each graph $\mathcal{G}_i$ is associated with a ground truth label $y_i^{\mathcal{G}}$ sampled from a label set $\hat{\mathcal{L}}$. Given a subset of labeled graphs (from a label set $\hat{\mathcal{L}}$), the goal is to learn a mapping $f^{\mathcal{G}} : (\mathbf{A}, \mathbf{X}^{(0)})_i \mapsto y_i^{\mathcal{G}}$ between each graph $\mathcal{G}_i$ and its ground truth label $y_i^{\mathcal{G}} \in \hat{\mathcal{L}}$. The graph classification loss is given by $L = \frac{1}{|\mathcal{G}_{\text{train}}|} \sum_{\mathcal{G}_i \in \mathcal{G}_{\text{train}}} \texttt{CrossEntropy} (\mathbf{x}^{\mathcal{G}_i}, y_i^{\mathcal{G}})$, where $\mathcal{G}_{\text{train}}$ is the training graph set and $\mathbf{x}^{\mathcal{G}_i}$ is the representation of graph $\mathcal{G}_i$.

**Evaluation of Size Generalizability.** Following prior work (6; 38), we evaluate the size generalizability of GNNs by testing their classification performance on graphs whose sizes are larger than those in the train set. We obtain the small training graphs and large test graphs from the same dataset.

## 2.3 Graph Neural Networks

GNNs can be designed from either the spatial perspective or the spectral perspective. Despite the difference in the design perspectives, a recent work (1) has shown that spectral GNNs and spatial GNNs are related and that spectral analysis of GNNs' behavior can provide a complementary point of view to understand GNNs in general. Most spatial GNNs (17; 34; 28; 12) use the message passing framework (11), which consists of three steps: neighborhood propagation, message combination and global pooling. Spectral GNNs (5; 10; 22) utilize the spectral properties of a propagation matrix $\mathbf{T}$ to perform the graph classification. The propagation matrix $\mathbf{T}$ is usually a function of the adjacency matrix $\mathbf{A}$, such as the normalized adjacency matrix $\mathbf{T} = (\mathbf{D}+\mathbf{I})^{-1/2}(\mathbf{A}+\mathbf{I})(\mathbf{D}+\mathbf{I})^{-1/2}$, or the normalized graph Laplacian matrix $\hat{\mathbf{L}}$. Since we consider an undirected graph with a real and symmetric adjacency matrix, the propagation matrix $\mathbf{T}$ is also real and symmetric. Then, we can perform the eigendecomposition on the propagation matrix $\mathbf{T}$: $\mathbf{T} = \mathbf{U}\mathbf{\Lambda}\mathbf{U}^T$, where $\mathbf{U}$ is an orthogonal matrix whose columns $\mathbf{U}_i$ are orthonormal and are the eigenvectors of $\mathbf{T}$, and $\mathbf{\Lambda}$ is a matrix whose diagonal elements are the eigenvalues of $\mathbf{T}$, sorted from large to small by their absolute values. The set of eigenvectors $\{\mathbf{U}_i\}$ form the orthonormal basis of $\mathbb{R}^n$. The goal of a spectral GNN is to learn a proper spectral filter: $f(\mathbf{\Lambda}) = c_0\mathbf{I} + c_1\mathbf{\Lambda} + c_2\mathbf{\Lambda}^2 + \cdots + c_i\mathbf{\Lambda}^i + \cdots$, where $c_i$ are the learnable coefficients. The convolution at each layer can be viewed as or is equivalent to: $\mathbf{X}^{(l+1)} = \sigma(\mathbf{U}f(\mathbf{\Lambda})\mathbf{U}^T\mathbf{X}^{(l)}\mathbf{W}^{(l)})$, where $\mathbf{W}^{(l)}$ is a learnable weight matrix, and $\sigma(\cdot)$ is a nonlinear function (e.g., ReLU). The graph representation is obtained from the node representations at the last convolution layer: $\mathbf{x}^{\mathcal{G}} = \text{Pooling}(\{\mathbf{x}_i^{(\text{Last})}\})$, where the Pooling function is performed on the set of all the node representations, and it can be Global_mean or Global_max or other more complex pooling functions (39; 18).

## 3 Spectral analysis of size-induced Distribution Shifts

In this section, we first show that the independence of the eigenvalue distribution of the propagation matrix $\mathbf{T}$ from the graph size is the key to achieving size generalizability of GNNs (§ 3.1). Next, focusing on biologically data, we empirically verify that the eigenvalue distribution of the propagation matrix depends on the graph size (§ 3.2). Finally, we explore the subgraph patterns responsible for the spectral disparities between small and large graphs, unveiling two key findings in § 3.3:

- Breaking cycles in graphs amplifies the spectrum difference between smaller and larger graphs.
- Extending cycle lengths in smaller graphs to match larger ones reduces the spectrum difference.

## 3.1 Graph spectrum and size generalizability of GNNs

To understand how GNNs generalize over graphs with different sizes, we examine the formulation of graph representations. In the context of spectral GNNs, graph representations rely on the eigenvalues of the propagation matrix. Consequently, the connection between graph representations and graph size reduces to the connection between the graph's spectrum and its size. More formally, we theoretically show the following proposition in Appendix A.

**Proposition 1** *When graphs of various sizes exhibit distinct eigenvalue distributions for the propagation matrix, the representations learned by spectral GNNs correlate with the graph size.*

The proposition suggests that for GNNs to achieve effective generalization to larger graphs, the disparity in the spectrum between small and large graphs should be small.

## 3.2 Size-related Spectrum Differences in Real-world Data

We now investigate how the eigenvalue distribution of the normalized adjacency matrix varies with graph size in real-world data. As indicated in Proposition 1, the spectrum discrepancy between small and large graphs affects the size generalizability of GNNs.

**Datasets.** We explore five pre-processed biological datasets (BBBP, BACE, NCI1, NCI109, and PROTEINS) from the Open Graph Benchmark (14) and TuDataset (23). More details about the datasets are provided in Appendix B.

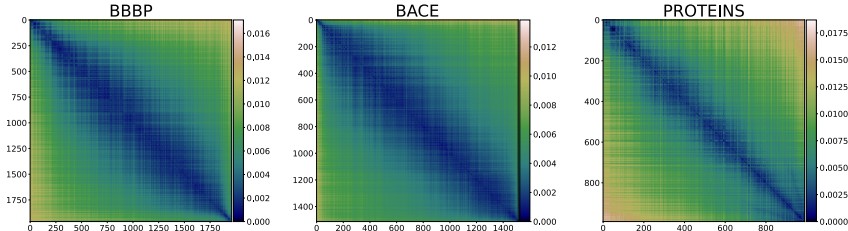

Figure 1: Pairwise graph distance of eigenvalue distributions: Graphs are sorted from small to large, and the (i,j)-th pixel in the plot represents the Wasserstein distance between the i-th and j-th graphs. Dark blue represents a small distance (high similarity), while light red represents a large distance (low similarity). We find that **eigenvalue distributions show a strong correlation with the graph size**.

Table 1: Average Wasserstein distance between graphs of 'similar sizes' and graphs of 'different sizes' based on **eigenvalue** distributions, respectively. The relative difference is computed by the difference of the Wasserstein distance normalized by the Wasserstein distance of similar graphs.

|  | **BBBP** | **BACE** | **PROTEINS** | **NCI109** | **NCI1** |
|---|---|---|---|---|---|
| **Different Size** | 0.00566 | 0.00411 | 0.00765 | 0.00563 | 0.00566 |
| **Similar Size** | 0.00184 | 0.00149 | 0.00261 | 0.00215 | 0.00215 |
| **Relative Difference** | 208% | 177% | 193% | 162% | 164% |

**Setup.** Figure 1 illustrates the pairwise distances of the graphs arranged in ascending order of size, where the distances are calculated using the Wasserstein distance (31). We represent the graphs by their empirical distributions of the eigenvalues that are obtained from the normalized adjacency matrix as suggested in (17): $\mathbf{T} = (\mathbf{D} + \mathbf{I})^{-1/2}(\mathbf{A} + \mathbf{I})(\mathbf{D} + \mathbf{I})^{-1/2}$. Using the normalized Laplacian matrix leads to similar observations. We note that the eigenvalues do not scale with the graph size, and they are bounded between [-1,1]. Dark blue represents a small distance (high similarity) while light red represents a large distance (low similarity).

**Results.** As can be seen in the three subplots in Figure 1, there is a wide blue band along the diagonal, which indicates that graphs of similar size have more similar eigenvalue distributions than graphs of different sizes. This suggests a strong correlation between the eigenvalue distributions and the graph size. To verify the observation quantitatively, we compute the distance of graphs with 'similar size' and graphs of 'different sizes' in Table 1. For each graph, we consider the 20 most 'similar graphs' in terms of size, and treat the remaining graphs as graphs of 'different sizes'. The table shows that the Wasserstein distances of eigenvalue distributions between the graphs of different sizes are significantly larger than the distances between graphs of similar size. Based on the empirical results and Proposition 1, the correlation between the eigenvalue distributions and the graph size results in the correlation of the final graph representation and the graph size, which prevents GNNs from generalizing over larger size.

### 3.3 Key Findings: Size-related Differences in Subgraph Patterns

In this subsection, we aim to identify the subgraph patterns that explain the spectrum differences between small and large graphs. Our empirical analysis pinpointed several peaks in the graph spectrum that match the spectrum of cycles. This motivated us to examine how cycle properties differ in small and large graphs and how these differences are revealed in the spectrum. Specifically, we aim to answer two questions: (Q1) How does the existence of cycles in the graphs influence the spectrum differences? (Q2) How do the variations in cycle lengths contribute to differences in the spectrum? In our analysis, we investigate the properties of the cycles in the cycle basis of each graph.

#### 3.3.1 Existence of Cycles & Spectrum: The Impact of Breaking Cycles

To understand how the existence of cycles in the graphs influences the spectrum differences, we break all basis cycles with minimal edge removal while maintaining the same number of disconnected components, according to the details and algorithm given in Appendix D.1. We analyze the impact of breaking cycles by assessing the corresponding changes in the spectrum. By following the convention

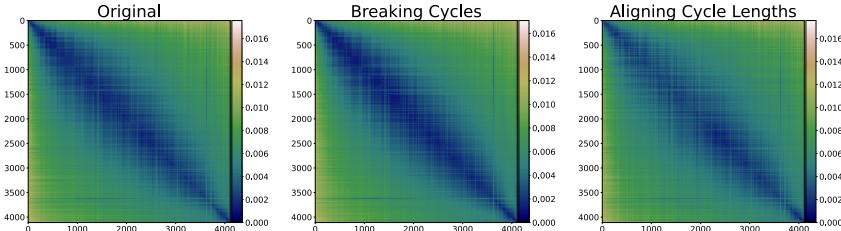

Figure 2: Pairwise graph distance measured by the Wasserstein distance of eigenvalue distributions after breaking cycles and aligning cycle lengths on the NCI109 dataset. Breaking cycles **amplifies the correlation** between eigenvalue distribution and graph size, while aligning cycle lengths **reduces the correlation**.

Table 2: Average Wasserstein distance of eigenvalue distributions between graphs of similar size and graphs of different sizes after breaking cycles and aligning cycle lengths. Relative difference is computed as in Table 1. We use ↑ (↓) to denote the increase (decrease) in the relative difference compared to not taking the corresponding action. Breaking cycles results in a larger relative difference, while aligning cycle lengths reduces the relative difference.

| | Breaking cycles | | | Aligning cycle lengths | | |
|---|---|---|---|---|---|---|
| **Datasets** | **Different sizes** | **Similar size** | **Δ relative difference** | **Different sizes** | **Similar size** | **Δ relative difference** |
| **BBBP** | 0.00545 | 0.00152 | ↑ 50% | 0.00565 | 0.00211 | ↓ 41% |
| **BACE** | 0.00420 | 0.00148 | ↑ 6% | 0.00417 | 0.00176 | ↓ 41% |
| **NCI1** | 0.00547 | 0.00173 | ↑ 53% | 0.00566 | 0.00242 | ↓ 31% |
| **NCI109** | 0.00548 | 0.00174 | ↑ 52% | 0.00568 | 0.00245 | ↓ 31% |
| **PROTEINS** | 0.00670 | 0.00212 | ↑ 31% | 0.00763 | 0.00302 | ↓ 41% |

of Section 3.2, in the center of Figure 2, we plot the pairwise graph distance based on eigenvalue distributions of graphs with different sizes after breaking cycles. The blue band along the diagonal of the plot becomes darker and narrower, suggesting a larger spectrum difference between small and large graphs and a stronger correlation between the spectrum and graph size. To evaluate the effects quantitatively, we further compute the changes in the relative spectrum difference and present the results in Table 2. These results indicate that failing to consider cycle information can lead to more significant differences in the spectrum between graphs of varying sizes, potentially causing GNNs to struggle with generalizing effectively to larger graphs.

### 3.3.2 CYCLE LENGTH & SPECTRUM: ALIGNING CYCLE LENGTHS

In Section 3.3.1, we showed that cycle information is crucial for GNNs to achieve size generalizability. We now further explore what cycle information helps reduce the spectrum difference between small and large graphs. To facilitate our exploration, we divide each real-world dataset into two subsets: one subset contains small graphs, and the other subset containts graphs of significantly larger size. Further details regarding this dataset split can be found in Appendix B. Using this dataset split, we observe a significant difference in the cycle lengths for small and large graphs (Appendix D.1). As described in Appendix D.1, to reduce that difference, we align the average cycle lengths between small and large graphs by randomly inserting redundant nodes to increase the cycle lengths in small graphs. The rightmost heatmap in Figure 2 shows how the correlation of eigenvalue distributions and graph size changes after aligning cycle lengths. We observe a lighter blue band along the diagonal, which suggests a weaker correlation between the spectrum and graph size. Furthermore, Table 2 quantitatively presents the changes in the relative spectrum difference between small and large graphs. We observe that aligning cycle lengths results in reduced disparities in the spectrum between graphs of different sizes. This indicates that GNNs capable of generalizing across varying cycle lengths may exhibit better size generalizability. In Appendix D.2, we show that our approach of aligning the cycle lengths is more effective at reducing the spectrum disparities than randomly adding the same number of nodes and edges.

## 4 METHODOLOGY: PROPOSED MODEL-AGNOSTIC STRATEGIES FOR GNNS

Our findings in Section 3 suggest that GNNs with better ability to identify cycles and generalize over cycle lengths may have better size generalizability on biological graphs. However, recent work ([8]) has found that most GNNs are incapable of learning cycle information. Inspired by these, we propose three model-agnostic strategies to help GNNs learn the cycle information.

### 4.1 STRATEGY 1: SIZE-INSENSITIVE ATTENTION

One way to incorporate cycle information into GNNs is by encoding it in the features and leveraging them within the attention mechanism to guide the learning process. Specifically, for each graph $\mathcal{G}$, we obtain its cycle basis $\mathcal{C}$. Then, for each node $v_i \in \mathcal{G}$, we calculate the average length of the cycle basis to which it belongs:

$$\ell_i = \begin{cases} \frac{\sum_{j=1}^{|\mathcal{C}|} |\mathcal{C}_j| \cdot \mathbb{1}_{\{v_i \in \mathcal{C}_j\}}}{\sum_{j=1}^{|\mathcal{C}|} \mathbb{1}_{\{v_i \in \mathcal{C}_j\}}} & \text{if } v_i \text{ belongs to some cycles} \\ 0 & \text{otherwise,} \end{cases} \tag{1}$$

where $\mathbb{1}_{\{\text{condition}\}}$ is an indicator function that outputs 1 when the condition is met and 0 otherwise. Then we manually construct a two-dimensional feature vector for each node $v_i$ based on its associated cycle information:

$$\mathbf{c}_i = [\mathbb{1}_{\{v_i \in \text{cycle}\}}, \ell_i]. \tag{2}$$

We use the structural feature matrix $\mathbf{C} = [\mathbf{c}_1; \ldots; \mathbf{c}_N] \in \mathbb{R}^{N \times 2}$ for attention. Since attention weights often diminish with increasing graph size due to the utilization of `Softmax`, we propose scaling the attention weights by the graph size and employing `Global_max` as the global pooling operation to mitigate the impact of graph size. Mathematically, our final graph representation is given by:

$$\mathbf{k} = \texttt{Softmax}(\mathbf{C}\mathbf{w}_A^\top) \cdot N, \quad \mathbf{x}^{\mathcal{G}} = \texttt{Global\_max}(\texttt{Diag}(\mathbf{k}) \cdot \mathbf{X}^{(\text{Last})}), \tag{3}$$

where $\mathbf{w}_A^\top$ is a learnable vector, and $\texttt{Diag}(\cdot)$ creates a diagonal matrix using the vector as its elements. We note that when we train on small graphs and test on large graphs, some structural features may not be seen in the training, such as certain cycle lengths in the large graphs. We rely on the attention mechanism to generalize to those cases.

### 4.2 STRATEGY 2: SELF-SUPERVISED AUXILIARY TASK

Our second proposed strategy utilizes a self-supervised auxiliary task to enhance the node representations with cycle-related information. The auxiliary task is to predict whether a node belongs to a cycle. We do not utilize cycle lengths as labels because large test graphs may have cycle lengths not present in the training data. Formally, let $X^{(\text{Last})}$ denote the node representations obtained after the last graph convolution. The conventional way of learning is to directly apply a pooling operator and then minimize the loss function for label supervision as below:

$$\mathcal{L}_{\text{label}} = \texttt{CrossEntropy}(\texttt{Linear}(\texttt{Pooling}(\mathbf{X}^{(\text{Last})}), y^{\mathcal{G}})), \tag{4}$$

where $y^{\mathcal{G}}$ is the ground truth label for the graph. In this approach, we incorporate an additional loss that aims to diffuse cycle-related information into the node representations through supervision:

$$\mathcal{L}_{\text{cycle}} = \texttt{CrossEntropy}(\texttt{MLP}(\mathbf{X}^{(\text{Last})}), \mathbf{y}_{\text{cycle}}), \tag{5}$$

where $\mathbf{y}_{\text{cycle}}$ is an indicator vector denoting whether a node belongs to a cycle. To sum up, the total loss is given by:

$$\mathcal{L} = \mathcal{L}_{\text{label}} + \lambda \mathcal{L}_{\text{cycle}}, \tag{6}$$

where $\lambda$ is a hyperparameter tuned via cross-validation.

### 4.3 STRATEGY 3: AUGMENTATION

This approach aims to reduce the discrepancy between small and large graphs through direct augmentation for the small training graphs. We augment the training graphs by extending the cycle lengths such that the average cycle length and standard deviation align with those in large graphs. To achieve this augmentation, we use the algorithm detailed in Appendix D.1. Additionally, we replicate the features from the nodes with the lowest degrees in the same cycle to populate the features for the newly added nodes. Last, we feed the augmented training graphs to GNNs for graph classification.

## 5 EXPERIMENTS

In this section, we conduct extensive experiments to evaluate our proposed strategies. We aim to answer the following research questions: (**RQ1**) **Effectiveness of cycle-aware strategies**: Do cycle-aware strategies effectively enhance the size generalizability of GNNs? And if so, which one proves to be more effective? (**RQ2**) **Comparison with other baselines**: How does the most efficient cycle-aware strategy compare to other baseline strategies?

### 5.1 SETUP

**Dataset.** We use the same biological datasets as in Section 3.2.

**Data Preprocessing and Important Training Details.** In order to analyze size generalizability, we have four splits for each dataset: train, validation, small_test, and large_test, where large_test contains graphs with significantly larger sizes. We generate the splits as follows. First, we sort the samples in the dataset by their size. Next, We take the train, validation, and small_test split from the 50% smallest graphs in the dataset. An intuitive way of getting the large_split is to take the top k largest graphs. However, doing so would result in severe label skewness (class imbalances) between the small_test and the large_test as demonstrated by Table 5 in Appendix C. To avoid such a severe label shift, we select the same number of graphs per class as in the small_test subset, starting from the largest graph within each class. This way guarantees that the label distribution between small_test and large_test is the same, while ensuring that the graph size in the latter is 2-10 times larger. Nevertheless, the smallest 50% samples still have significant **class imbalance**. To address this issue, we use upsampling during training throughout the experiments, and we use **F1** as the metric to measure the model performance. More details about data preprocessing, hyperparameters, and training can be found in Appendix C and Appendix F.

**Baselines.** We use six neural network models as our GNN backbones. Each model consists of three layers, with a global max pooling layer employed in the final layer. The baseline models are: Multilayer Perceptron (MLP), GCN (17), GAT (28), GIN (34), FAGCN (5), and GNNML3 (2). We integrate six model-agnostic strategies with these GNN backbones. For our three proposed strategies, we use (1) +SSL to denote the use of self-supervised auxiliary task, (2) +AugCyc to denote the use of cycle-length augmentation, and (3) +SIA to denote the use of structural-based size-insensitive attention. We also compare with other model-agnostic strategies: (4) thresholded SAG pooling (18; 20) (+SAGPool), an attention-based pooling method effective for generalizing to large and noisy graphs; (5) SizeShiftReg (6) (+SSR), a regularization based on the idea of simulating a shift in the size of the training graphs using coarsening techniques; (6) RPGNN (24) (+RPGNN), an expressive model for arbitrary-sized graphs; (7) two versions of CIGA (+CIGAv1 & +CIGAv2) (4), a causal model good at handling out-of-distribution problems on graphs. Besides these model-agnostic strategies, our baselines also include an expressive model SMP (30) (SMP), which excels at the cycle detection task.

### 5.2 (RQ1) EFFECTIVENESS OF CYCLE-AWARE STRATEGIES

In this section, we aim to evaluate the effectiveness of our proposed cycle-aware strategies in enhancing the size generalizability of GNNs and determine the most effective strategy.

As mentioned in Section 2.2, we evaluate the size generalizability of GNNs by training them on small graphs and testing their graph classification performance on large graphs. Better size generalizability translates into better performance on large graphs. Table 3 showcases the size generalizability results for our three proposed cycle-aware strategies, which we evaluate across five distinct datasets and six different backbone models. Notably, to better compare different strategies, the last column gives the average improvements compared with the original model evaluated across all datasets.

First, all of our proposed strategies consistently lead to improvements in large test datasets without sacrificing the performance on small graphs. On average, these enhancements can reach up to 8.4%, affirming the effectiveness of cycle information in improving GNN size generalizability, as discussed in Section 3. Second, it is worth noting that cycle lengths provide more valuable information for enhancing size generalizability of GNNs. This is evident from the consistently better performance of the strategies +AugCyc and +SIA compared to the +SSL strategy on large test graphs, which solely predicts whether a node belongs to a cycle in the auxiliary task. Third, the simple attention-based

Table 3: Size generalizability evaluated by the graph classification performance on small and large test graphs. The performance is reported by the average F1 scores and its standard deviation. The rightmost column denotes the average improvements compared with the original performance using the same backbone model across five different datasets. The largest average improvement within the same model and small/large category is highlighted in orange. **All strategies enhance GNNs' size generalizability, with `+SIA` surprisingly emerging as the most effective method.**

| Datasets | BBBP | | BACE | | PROTEINS | | NCI1 | | NCI109 | | Avg Improv | |
|---|---|---|---|---|---|---|---|---|---|---|---|---|
| Models | Small | Large | Small | Large | Small | Large | Small | Large | Small | Large | Small | Large |
| MLP | $90.36_{\pm0.71}$ | $55.61_{\pm3.37}$ | $61.06_{\pm5.79}$ | $21.06_{\pm7.89}$ | $36.15_{\pm2.28}$ | $21.55_{\pm1.34}$ | $36.43_{\pm3.89}$ | $3.36_{\pm2.87}$ | $35.87_{\pm4.23}$ | $4.65_{\pm3.72}$ | - | - |
| MLP+SSL | $90.90_{\pm1.76}$ | $62.56_{\pm5.48}$ | $58.57_{\pm8.85}$ | $23.01_{\pm11.95}$ | $35.00_{\pm2.8}$ | $20.88_{\pm1.64}$ | $34.71_{\pm1.33}$ | $2.86_{\pm0.78}$ | $37.29_{\pm4.69}$ | $6.34_{\pm4.78}$ | -0.68 | +1.88 |
| MLP+AugCyc | $90.72_{\pm2.91}$ | $57.86_{\pm4.74}$ | $59.88_{\pm7.33}$ | $26.50_{\pm14.97}$ | $37.29_{\pm0.0}$ | $22.22_{\pm0.0}$ | $36.98_{\pm2.29}$ | $2.64_{\pm2.1}$ | $40.59_{\pm3.86}$ | $8.41_{\pm3.71}$ | +1.12 | +2.28 |
| MLP+SIA | $90.38_{\pm1.05}$ | $62.79_{\pm7.55}$ | $60.85_{\pm7.83}$ | $21.79_{\pm15.07}$ | $40.68_{\pm3.56}$ | $33.57_{\pm11.87}$ | $35.42_{\pm3.83}$ | $3.26_{\pm2.55}$ | $39.20_{\pm2.96}$ | $12.2_{\pm5.05}$ | +1.33 | +5.48 |
| GCN | $91.37_{\pm0.59}$ | $68.59_{\pm7.47}$ | $63.68_{\pm6.63}$ | $28.72_{\pm14.26}$ | $72.35_{\pm2.58}$ | $40.57_{\pm7.67}$ | $54.91_{\pm2.37}$ | $28.80_{\pm7.57}$ | $60.83_{\pm1.92}$ | $30.45_{\pm4.34}$ | - | - |
| GCN+SSL | $92.66_{\pm1.21}$ | $73.24_{\pm5.71}$ | $64.92_{\pm4.44}$ | $32.84_{\pm16.08}$ | $72.46_{\pm1.58}$ | $41.21_{\pm6.66}$ | $57.43_{\pm3.23}$ | $32.58_{\pm10.08}$ | $60.50_{\pm3.09}$ | $27.35_{\pm11.42}$ | +0.97 | +2.01 |
| GCN+AugCyc | $91.41_{\pm1.33}$ | $68.08_{\pm7.65}$ | $63.83_{\pm5.44}$ | $35.65_{\pm7.70}$ | $72.87_{\pm3.68}$ | $54.73_{\pm8.24}$ | $53.85_{\pm3.71}$ | $27.39_{\pm8.33}$ | $62.78_{\pm2.98}$ | $33.62_{\pm3.58}$ | +0.32 | +4.47 |
| GCN+SIA | $91.32_{\pm0.73}$ | $71.66_{\pm6.99}$ | $64.35_{\pm9.76}$ | $24.24_{\pm17.03}$ | $73.84_{\pm3.65}$ | $58.74_{\pm9.49}$ | $59.78_{\pm1.65}$ | $45.70_{\pm6.70}$ | $60.32_{\pm2.90}$ | $38.78_{\pm4.55}$ | +1.29 | +8.40 |
| GAT | $91.27_{\pm1.43}$ | $68.35_{\pm7.02}$ | $69.73_{\pm2.05}$ | $42.23_{\pm11.18}$ | $72.25_{\pm4.25}$ | $43.86_{\pm6.82}$ | $58.22_{\pm2.86}$ | $49.36_{\pm4.12}$ | $64.39_{\pm3.29}$ | $38.36_{\pm8.93}$ | - | - |
| GAT+SSL | $91.65_{\pm0.92}$ | $74.24_{\pm7.34}$ | $71.20_{\pm2.04}$ | $40.88_{\pm10.81}$ | $74.20_{\pm1.46}$ | $49.30_{\pm5.56}$ | $59.47_{\pm2.89}$ | $51.85_{\pm4.03}$ | $66.79_{\pm3.56}$ | $42.20_{\pm6.71}$ | +1.49 | +3.26 |
| GAT+AugCyc | $92.41_{\pm1.29}$ | $69.57_{\pm2.89}$ | $68.39_{\pm6.06}$ | $40.73_{\pm13.4}$ | $74.99_{\pm1.89}$ | $59.80_{\pm7.27}$ | $56.23_{\pm3.85}$ | $49.37_{\pm7.52}$ | $64.07_{\pm3.46}$ | $45.25_{\pm9.19}$ | +0.05 | +4.51 |
| GAT+SIA | $91.88_{\pm2.12}$ | $74.87_{\pm5.62}$ | $69.64_{\pm6.79}$ | $43.87_{\pm7.98}$ | $75.35_{\pm3.28}$ | $62.71_{\pm4.98}$ | $61.42_{\pm1.07}$ | $55.73_{\pm12.98}$ | $63.27_{\pm3.15}$ | $45.97_{\pm7.74}$ | +1.14 | +8.20 |
| GIN | $88.28_{\pm2.39}$ | $66.67_{\pm5.55}$ | $57.02_{\pm6.48}$ | $22.97_{\pm10.26}$ | $74.55_{\pm4.27}$ | $50.20_{\pm5.36}$ | $62.17_{\pm3.86}$ | $44.26_{\pm7.03}$ | $62.42_{\pm2.77}$ | $33.23_{\pm6.77}$ | - | - |
| GIN+SSL | $91.13_{\pm1.32}$ | $68.67_{\pm9.75}$ | $56.46_{\pm8.59}$ | $23.91_{\pm10.64}$ | $75.47_{\pm1.15}$ | $48.14_{\pm4.00}$ | $61.18_{\pm3.53}$ | $46.47_{\pm9.86}$ | $63.11_{\pm4.05}$ | $35.0_{\pm11.43}$ | +0.58 | +0.97 |
| GIN+AugCyc | $92.56_{\pm1.17}$ | $77.69_{\pm5.63}$ | $58.30_{\pm5.29}$ | $23.89_{\pm13.17}$ | $74.56_{\pm2.92}$ | $51.02_{\pm8.42}$ | $62.70_{\pm0.94}$ | $46.76_{\pm5.34}$ | $64.56_{\pm5.45}$ | $37.16_{\pm5.86}$ | +1.65 | +3.84 |
| GIN+SIA | $92.70_{\pm0.45}$ | $75.99_{\pm4.74}$ | $61.30_{\pm6.77}$ | $24.42_{\pm16.37}$ | $74.88_{\pm4.24}$ | $51.36_{\pm7.76}$ | $62.83_{\pm1.07}$ | $42.82_{\pm8.92}$ | $63.00_{\pm4.24}$ | $41.65_{\pm4.19}$ | +2.05 | +3.78 |
| FAGCN | $90.58_{\pm1.72}$ | $64.93_{\pm7.62}$ | $62.96_{\pm2.12}$ | $24.65_{\pm11.71}$ | $70.03_{\pm5.20}$ | $42.34_{\pm6.61}$ | $43.51_{\pm4.29}$ | $10.16_{\pm7.80}$ | $55.78_{\pm3.5}$ | $22.65_{\pm12.87}$ | - | - |
| FAGCN+SSL | $91.55_{\pm2.51}$ | $67.56_{\pm5.48}$ | $64.67_{\pm3.88}$ | $35.46_{\pm16.52}$ | $66.97_{\pm1.75}$ | $48.06_{\pm8.33}$ | $46.42_{\pm6.08}$ | $12.11_{\pm5.39}$ | $56.04_{\pm4.29}$ | $23.99_{\pm10.57}$ | +0.56 | +4.49 |
| FAGCN+AugCyc | $91.30_{\pm2.26}$ | $71.44_{\pm6.45}$ | $57.68_{\pm3.38}$ | $26.41_{\pm23.39}$ | $68.85_{\pm16.12}$ | $44.39_{\pm16.89}$ | $39.48_{\pm4.99}$ | $10.98_{\pm5.45}$ | $55.30_{\pm3.46}$ | $24.59_{\pm9.19}$ | -2.05 | +2.62 |
| FAGCN+SIA | $90.17_{\pm2.83}$ | $74.65_{\pm9.13}$ | $62.40_{\pm3.36}$ | $30.35_{\pm13.48}$ | $71.30_{\pm5.79}$ | $48.94_{\pm10.62}$ | $46.95_{\pm5.71}$ | $10.99_{\pm7.50}$ | $52.82_{\pm6.28}$ | $19.08_{\pm5.32}$ | +0.16 | +3.86 |
| GNNML3 | $92.01_{\pm1.56}$ | $64.18_{\pm6.99}$ | $62.31_{\pm4.90}$ | $32.94_{\pm12.86}$ | $71.59_{\pm3.5}$ | $40.74_{\pm15.0}$ | $63.73_{\pm4.67}$ | $51.75_{\pm9.05}$ | $59.39_{\pm3.76}$ | $33.80_{\pm11.19}$ | - | - |
| GNNML3+SSL | $92.96_{\pm1.54}$ | $64.18_{\pm8.62}$ | $65.65_{\pm5.69}$ | $31.78_{\pm12.72}$ | $74.41_{\pm3.21}$ | $56.81_{\pm3.49}$ | $63.91_{\pm3.34}$ | $48.84_{\pm10.07}$ | $61.01_{\pm2.44}$ | $35.13_{\pm9.49}$ | +1.78 | +2.67 |
| GNNML3+AugCyc | $91.38_{\pm2.92}$ | $69.82_{\pm5.51}$ | $63.36_{\pm2.78}$ | $32.59_{\pm10.32}$ | $70.54_{\pm5.00}$ | $38.79_{\pm5.21}$ | $62.30_{\pm3.27}$ | $55.57_{\pm11.73}$ | $58.18_{\pm3.17}$ | $41.30_{\pm14.75}$ | -0.65 | +2.93 |
| GNNML3+SIA | $92.70_{\pm0.81}$ | $70.43_{\pm6.36}$ | $64.57_{\pm2.72}$ | $37.73_{\pm7.68}$ | $69.32_{\pm3.79}$ | $48.94_{\pm10.62}$ | $63.91_{\pm5.81}$ | $48.85_{\pm12.11}$ | $61.58_{\pm3.98}$ | $49.70_{\pm17.85}$ | +0.61 | +6.45 |

strategy `+SIA` achieves the best overall performance improvements in all scenarios. On average, the `+SIA` strategy enhances both in-distribution and out-of-distribution generalization, while `+SSL` and `+AugCyc` excel particularly in out-of-distribution generalization. Additionally, `+SIA` achieves the highest average improvements on large test graphs. We attribute this to the challenges GNNs face in effectively learning cycle information, as shown in recent literature (8). Furthermore, in Appendix E, we conduct an ablation study further demonstrating that the attention mechanism, without considering the cycle information, cannot improve the size generalizability of GNNs.

### 5.3 (RQ2) COMPARISON WITH OTHER BASELINES

We now compare our best-performing strategy, `+SIA`, with other approaches and present their respective graph classification performances in Table 3. We find that `+SIA` consistently achieves the best performance compared with other baseline methods. While `+SAGPool`, `+RPGNN`, `+SSR`, and `+CIGA` also enhance the size generalizability of GNNs, the improvements are less pronounced than those of `+SIA`. Additionally, it's worth noting that `+SAGPool` and `+CIGAv1` show sensitivity to hyperparameters. While not explicitly designed for size generalizability, the expressive model `SMP` demonstrates strong performance on the BBBP dataset due to its cycle detection ability, validating our empirical insights.

## 6 RELATED WORK

Literature on GNNs' size generalization presents conflicting views. Empirical studies highlight GNNs' size generalizability in physics simulation (26) and algorithmic reasoning (35). Levie et al. (21) theoretically showed that spectral GNNs robustly transfer between graphs with varied sizes when discretizing the same underlying space. Meanwhile, some arguments suggest that GNNs may require additional assistance to achieve size generalizability. For instance, Yan et al. (36) and Velivckovic et al. (29) found that neural networks effectively generalize to larger graphs than those used in training when attention weights are properly supervised. Conversely, some argue that GNN performance degrades with size shifts between training and test data, leading to the proposal of various models to mitigate this challenge. For instance, Yehudai et al. (38) argued that this performance degradation can be attributed to the changes in the local degree patterns. Knyazev et al. (18) found that using attention with proper thresholding can improve the size generalizability of GNNs. Buffelli et al. (6) simulated a size shift in the training graphs via graph coarsening and proposed a regularization that makes the

Table 4: Size generalizability evaluated with other baselines, following the same rule as in Table 3. `+SIA` consistently and significantly outperforms other strategies regarding size generalizability.

| Datasets | BBBP | | BACE | | PROTEINS | | NCI1 | | NCI109 | | Avg Improv | |
|---|---|---|---|---|---|---|---|---|---|---|---|---|
| Models | Small | Large | Small | Large | Small | Large | Small | Large | Small | Large | Small | Large |
| MLP | $90.36_{\pm0.71}$ | $55.61_{\pm3.37}$ | $61.06_{\pm5.79}$ | $21.06_{\pm7.89}$ | $36.15_{\pm2.28}$ | $21.55_{\pm1.34}$ | $36.43_{\pm3.89}$ | $3.36_{\pm2.87}$ | $35.87_{\pm4.23}$ | $4.65_{\pm3.72}$ | - | - |
| MLP+SAGPool | $89.42_{\pm5.84}$ | $52.52_{\pm4.98}$ | $65.82_{\pm9.42}$ | $13.04_{\pm5.02}$ | $42.54_{\pm12.82}$ | $20.84_{\pm7.84}$ | $45.41_{\pm13.21}$ | $12.32_{\pm15.78}$ | $42.18_{\pm9.88}$ | $15.35_{\pm12.31}$ | +5.10 | +1.57 |
| MLP+RPGNN | $90.44_{\pm1.03}$ | $55.10_{\pm7.64}$ | $57.80_{\pm8.53}$ | $21.26_{\pm8.20}$ | $45.60_{\pm2.69}$ | $20.71_{\pm1.41}$ | $35.34_{\pm4.35}$ | $13.45_{\pm25.12}$ | $38.60_{\pm2.37}$ | $10.44_{\pm18.77}$ | +1.58 | +2.95 |
| MLP+SSR | $91.07_{\pm0.67}$ | $57.02_{\pm9.04}$ | $60.35_{\pm5.36}$ | $25.42_{\pm4.02}$ | $37.15_{\pm2.48}$ | $22.77_{\pm4.28}$ | $34.42_{\pm1.89}$ | $1.27_{\pm0.63}$ | $38.68_{\pm4.70}$ | $6.96_{\pm4.65}$ | +0.36 | +1.44 |
| MLP+CIGAv1 | $88.15_{\pm1.05}$ | $54.14_{\pm6.07}$ | $58.18_{\pm10.81}$ | $24.99_{\pm14.41}$ | $34.68_{\pm4.23}$ | $18.23_{\pm2.50}$ | $33.52_{\pm7.85}$ | $8.93_{\pm7.48}$ | $32.79_{\pm1.20}$ | $3.19_{\pm0.01}$ | -2.51 | +0.65 |
| MLP+CIGAv2 | $88.08_{\pm3.84}$ | $61.70_{\pm13.15}$ | $57.46_{\pm4.71}$ | $18.43_{\pm16.73}$ | $38.28_{\pm4.23}$ | $24.87_{\pm5.62}$ | $35.77_{\pm7.55}$ | $4.22_{\pm6.89}$ | $38.45_{\pm3.68}$ | $4.55_{\pm1.28}$ | -0.37 | +1.51 |
| MLP+SIA | $90.38_{\pm1.05}$ | $62.79_{\pm7.55}$ | $60.85_{\pm7.83}$ | $21.79_{\pm15.07}$ | $40.68_{\pm3.56}$ | $33.57_{\pm11.87}$ | $35.42_{\pm3.83}$ | $3.26_{\pm2.55}$ | $39.20_{\pm2.96}$ | $12.2_{\pm5.05}$ | +1.33 | +5.48 |
| GCN | $91.37_{\pm0.59}$ | $68.59_{\pm7.47}$ | $63.68_{\pm6.63}$ | $28.72_{\pm14.26}$ | $72.35_{\pm2.58}$ | $40.57_{\pm7.67}$ | $54.91_{\pm2.37}$ | $28.80_{\pm7.57}$ | $60.83_{\pm1.92}$ | $30.45_{\pm4.34}$ | - | - |
| GCN+SAGPool | $92.05_{\pm3.95}$ | $67.06_{\pm5.18}$ | $57.59_{\pm6.65}$ | $42.74_{\pm14.7}$ | $68.75_{\pm5.09}$ | $32.98_{\pm3.26}$ | $58.56_{\pm8.30}$ | $38.73_{\pm20.94}$ | $62.87_{\pm16.7}$ | $30.93_{\pm10.59}$ | -0.66 | +3.06 |
| GCN+RPGNN | $92.27_{\pm0.81}$ | $68.69_{\pm6.58}$ | $63.70_{\pm1.06}$ | $33.86_{\pm13.91}$ | $74.74_{\pm3.75}$ | $24.61_{\pm10.08}$ | $58.88_{\pm2.03}$ | $34.68_{\pm10.77}$ | $63.10_{\pm1.86}$ | $39.69_{\pm5.88}$ | +1.91 | +0.88 |
| GCN+SSR | $91.19_{\pm1.14}$ | $68.15_{\pm6.38}$ | $66.01_{\pm2.51}$ | $31.64_{\pm9.96}$ | $73.51_{\pm2.91}$ | $43.33_{\pm5.19}$ | $59.60_{\pm2.62}$ | $35.01_{\pm7.13}$ | $59.78_{\pm2.71}$ | $33.11_{\pm6.44}$ | +1.39 | +2.82 |
| GCN+CIGAv1 | $90.55_{\pm1.32}$ | $66.55_{\pm5.80}$ | $66.66_{\pm5.72}$ | $28.51_{\pm8.64}$ | $72.64_{\pm1.81}$ | $54.67_{\pm6.08}$ | $58.52_{\pm4.88}$ | $40.82_{\pm11.14}$ | $59.09_{\pm3.50}$ | $25.82_{\pm7.81}$ | +0.86 | +3.85 |
| GCN+CIGAv2 | $89.45_{\pm3.60}$ | $69.71_{\pm8.20}$ | $65.02_{\pm1.80}$ | $35.42_{\pm12.36}$ | $72.15_{\pm3.86}$ | $60.12_{\pm6.84}$ | $57.89_{\pm3.74}$ | $35.42_{\pm10.75}$ | $58.12_{\pm5.37}$ | $28.51_{\pm10.10}$ | -0.10 | +6.41 |
| GCN+SIA | $91.32_{\pm0.73}$ | $71.66_{\pm6.99}$ | $64.35_{\pm9.76}$ | $24.24_{\pm17.03}$ | $73.84_{\pm3.65}$ | $58.74_{\pm9.49}$ | $59.78_{\pm1.65}$ | $45.70_{\pm6.70}$ | $60.32_{\pm2.90}$ | $38.78_{\pm4.55}$ | +1.29 | +8.40 |
| GAT | $91.27_{\pm1.43}$ | $68.35_{\pm7.02}$ | $69.73_{\pm2.05}$ | $42.23_{\pm11.18}$ | $72.25_{\pm4.25}$ | $43.86_{\pm6.82}$ | $58.22_{\pm2.86}$ | $49.36_{\pm4.12}$ | $64.39_{\pm3.29}$ | $38.36_{\pm8.93}$ | - | - |
| GAT+SAGPool | $89.90_{\pm2.15}$ | $60.39_{\pm17.18}$ | $66.10_{\pm6.44}$ | $46.40_{\pm15.45}$ | $73.85_{\pm8.60}$ | $38.60_{\pm6.34}$ | $55.25_{\pm1.43}$ | $52.71_{\pm3.02}$ | $65.32_{\pm3.42}$ | $43.20_{\pm19.94}$ | -1.09 | -0.17 |
| GAT+RPGNN | $91.76_{\pm2.69}$ | $65.85_{\pm5.37}$ | $69.97_{\pm2.17}$ | $39.27_{\pm13.50}$ | $72.89_{\pm3.35}$ | $38.49_{\pm6.14}$ | $59.31_{\pm5.51}$ | $58.18_{\pm5.76}$ | $65.52_{\pm1.94}$ | $44.15_{\pm5.76}$ | -0.75 | +0.76 |
| GAT+SSR | $91.98_{\pm0.66}$ | $74.83_{\pm4.35}$ | $66.03_{\pm3.83}$ | $41.41_{\pm11.8}$ | $74.72_{\pm3.51}$ | $44.81_{\pm8.59}$ | $60.68_{\pm1.95}$ | $49.64_{\pm5.26}$ | $66.73_{\pm1.65}$ | $41.14_{\pm4.41}$ | +0.86 | +1.93 |
| GAT+CIGAv1 | $89.53_{\pm1.51}$ | $67.35_{\pm8.74}$ | $67.18_{\pm5.12}$ | $39.88_{\pm18.64}$ | $73.28_{\pm4.87}$ | $48.56_{\pm5.42}$ | $59.52_{\pm3.27}$ | $54.35_{\pm11.85}$ | $66.82_{\pm2.03}$ | $50.62_{\pm4.85}$ | +0.09 | +3.72 |
| GAT+CIGAv2 | $90.92_{\pm1.43}$ | $72.08_{\pm7.09}$ | $66.57_{\pm4.91}$ | $40.93_{\pm19.35}$ | $74.68_{\pm7.54}$ | $60.88_{\pm3.48}$ | $56.88_{\pm24.40}$ | $54.28_{\pm22.63}$ | $66.78_{\pm3.20}$ | $52.62_{\pm7.98}$ | -0.01 | +7.73 |
| GAT+SIA | $91.88_{\pm2.12}$ | $74.87_{\pm5.62}$ | $69.64_{\pm6.79}$ | $43.87_{\pm7.98}$ | $75.35_{\pm3.28}$ | $62.71_{\pm4.98}$ | $61.42_{\pm1.07}$ | $55.73_{\pm12.98}$ | $63.27_{\pm3.15}$ | $45.97_{\pm7.74}$ | +1.14 | +8.20 |
| GIN | $88.28_{\pm2.39}$ | $66.67_{\pm5.55}$ | $57.02_{\pm6.48}$ | $22.97_{\pm10.26}$ | $74.55_{\pm4.27}$ | $50.20_{\pm5.36}$ | $62.17_{\pm3.86}$ | $44.26_{\pm7.03}$ | $62.42_{\pm2.77}$ | $33.23_{\pm6.77}$ | - | - |
| GIN+SAGPool | $91.56_{\pm1.32}$ | $71.20_{\pm11.86}$ | $62.22_{\pm7.45}$ | $26.17_{\pm17.98}$ | $68.73_{\pm13.46}$ | $35.77_{\pm21.54}$ | $65.50_{\pm4.50}$ | $45.66_{\pm3.56}$ | $59.29_{\pm5.83}$ | $44.64_{\pm9.53}$ | +0.57 | +1.22 |
| GIN+RPGNN | $89.59_{\pm1.33}$ | $69.23_{\pm8.05}$ | $57.23_{\pm7.07}$ | $16.28_{\pm9.31}$ | $71.63_{\pm5.85}$ | $45.11_{\pm15.98}$ | $61.59_{\pm4.12}$ | $48.86_{\pm5.88}$ | $62.27_{\pm2.33}$ | $44.04_{\pm7.06}$ | -0.43 | +1.24 |
| GIN+SSR | $89.00_{\pm1.77}$ | $68.84_{\pm6.01}$ | $59.83_{\pm2.34}$ | $21.28_{\pm19.26}$ | $72.46_{\pm2.86}$ | $55.46_{\pm15.95}$ | $62.54_{\pm1.30}$ | $48.73_{\pm8.62}$ | $61.05_{\pm3.37}$ | $35.63_{\pm7.29}$ | +0.09 | +2.52 |
| GIN+CIGAv1 | $91.01_{\pm1.59}$ | $74.85_{\pm10.41}$ | $60.58_{\pm7.15}$ | $23.78_{\pm17.58}$ | $75.32_{\pm16.25}$ | $54.83_{\pm9.87}$ | $61.94_{\pm1.08}$ | $45.85_{\pm5.83}$ | $61.55_{\pm12.55}$ | $35.88_{\pm7.77}$ | +1.19 | +3.57 |
| GIN+CIGAv2 | $90.66_{\pm1.72}$ | $75.80_{\pm14.30}$ | $63.02_{\pm1.80}$ | $22.42_{\pm12.36}$ | $73.25_{\pm5.42}$ | $53.35_{\pm8.76}$ | $64.42_{\pm5.35}$ | $45.37_{\pm5.12}$ | $59.52_{\pm4.68}$ | $38.42_{\pm5.68}$ | +1.29 | +3.61 |
| GIN+SIA | $92.70_{\pm0.45}$ | $75.99_{\pm4.74}$ | $61.30_{\pm6.77}$ | $24.42_{\pm16.37}$ | $74.88_{\pm4.24}$ | $51.36_{\pm7.76}$ | $62.83_{\pm1.07}$ | $42.82_{\pm8.92}$ | $63.00_{\pm4.24}$ | $41.65_{\pm4.19}$ | +2.05 | +3.78 |
| FAGCN | $90.58_{\pm1.72}$ | $64.93_{\pm7.62}$ | $62.96_{\pm2.12}$ | $24.65_{\pm11.71}$ | $70.03_{\pm5.20}$ | $42.34_{\pm6.61}$ | $43.51_{\pm4.29}$ | $10.16_{\pm7.80}$ | $55.78_{\pm3.5}$ | $22.65_{\pm12.87}$ | - | - |
| FAGCN+SAGPool | $88.08_{\pm7.26}$ | $62.67_{\pm15.01}$ | $62.78_{\pm4.28}$ | $31.50_{\pm12.13}$ | $72.41_{\pm4.85}$ | $50.09_{\pm16.49}$ | $45.66_{\pm4.51}$ | $11.21_{\pm2.60}$ | $58.04_{\pm22.06}$ | $15.43_{\pm4.69}$ | +0.28 | +1.23 |
| FAGCN+RPGNN | $90.43_{\pm2.58}$ | $69.58_{\pm11.71}$ | $61.00_{\pm51.91}$ | $20.94_{\pm12.62}$ | $68.71_{\pm3.58}$ | $43.58_{\pm12.21}$ | $44.55_{\pm5.82}$ | $12.22_{\pm5.95}$ | $57.03_{\pm1.08}$ | $21.86_{\pm13.32}$ | -0.23 | +0.69 |
| FAGCN+SSR | $88.95_{\pm2.16}$ | $70.12_{\pm9.49}$ | $64.12_{\pm2.57}$ | $22.37_{\pm7.80}$ | $67.92_{\pm3.00}$ | $42.27_{\pm8.69}$ | $47.47_{\pm2.77}$ | $14.69_{\pm8.22}$ | $55.66_{\pm3.37}$ | $22.35_{\pm9.35}$ | +0.25 | +1.41 |
| FAGCN+CIGAv1 | $91.08_{\pm1.48}$ | $69.29_{\pm3.35}$ | $62.66_{\pm5.72}$ | $28.51_{\pm8.64}$ | $68.58_{\pm6.19}$ | $57.79_{\pm10.45}$ | $40.93_{\pm10.69}$ | $9.45_{\pm11.16}$ | $48.07_{\pm5.70}$ | $15.11_{\pm5.82}$ | +1.19 | +3.57 |
| FAGCN+CIGAv2 | $92.08_{\pm1.70}$ | $68.37_{\pm9.67}$ | $60.37_{\pm4.65}$ | $22.29_{\pm15.00}$ | $69.45_{\pm4.97}$ | $60.28_{\pm12.24}$ | $44.27_{\pm6.74}$ | $15.88_{\pm10.25}$ | $50.25_{\pm4.44}$ | $16.22_{\pm7.74}$ | -1.29 | +3.66 |
| FAGCN+SIA | $90.17_{\pm2.83}$ | $74.65_{\pm9.13}$ | $62.40_{\pm3.36}$ | $30.35_{\pm13.48}$ | $71.30_{\pm5.79}$ | $48.94_{\pm10.62}$ | $46.95_{\pm5.71}$ | $10.99_{\pm7.50}$ | $52.82_{\pm6.28}$ | $19.08_{\pm5.32}$ | +0.16 | +3.86 |
| GNNML3 | $92.01_{\pm1.56}$ | $64.18_{\pm6.99}$ | $62.31_{\pm4.90}$ | $32.94_{\pm12.86}$ | $71.59_{\pm3.5}$ | $40.74_{\pm15.0}$ | $63.73_{\pm4.67}$ | $51.75_{\pm9.05}$ | $59.39_{\pm3.76}$ | $33.80_{\pm11.19}$ | - | - |
| GNNML3+SAGPool | $89.62_{\pm3.84}$ | $63.25_{\pm27.87}$ | $59.35_{\pm7.27}$ | $37.30_{\pm12.49}$ | $65.67_{\pm7.11}$ | $34.79_{\pm20.12}$ | $65.34_{\pm1.94}$ | $54.29_{\pm3.58}$ | $60.38_{\pm7.23}$ | $46.61_{\pm15.23}$ | +0.14 | +2.73 |
| GNNML3+RPGNN | $92.57_{\pm1.45}$ | $72.30_{\pm9.54}$ | $61.85_{\pm3.67}$ | $27.54_{\pm12.03}$ | $70.48_{\pm2.46}$ | $38.61_{\pm14.61}$ | $64.60_{\pm2.14}$ | $50.25_{\pm8.65}$ | $60.22_{\pm2.40}$ | $36.88_{\pm15.93}$ | +0.14 | +0.43 |
| GNNML3+SSR | $91.86_{\pm1.30}$ | $69.96_{\pm5.56}$ | $64.95_{\pm2.82}$ | $26.56_{\pm4.68}$ | $74.33_{\pm1.85}$ | $49.02_{\pm7.42}$ | $63.19_{\pm4.35}$ | $54.30_{\pm10.33}$ | $63.27_{\pm2.74}$ | $45.74_{\pm11.47}$ | +1.41 | +4.43 |
| GNNML3+CIGAv1 | $91.25_{\pm4.67}$ | $72.80_{\pm5.88}$ | $64.23_{\pm2.10}$ | $34.95_{\pm15.92}$ | $73.89_{\pm4.73}$ | $52.01_{\pm12.59}$ | $61.89_{\pm4.53}$ | $45.25_{\pm7.80}$ | $60.95_{\pm3.44}$ | $38.80_{\pm12.36}$ | +0.64 | +4.08 |
| GNNML3+CIGAv2 | $89.61_{\pm1.46}$ | $67.17_{\pm5.94}$ | $64.19_{\pm5.36}$ | $27.28_{\pm12.79}$ | $75.07_{\pm2.64}$ | $54.50_{\pm9.29}$ | $60.28_{\pm4.56}$ | $46.25_{\pm10.42}$ | $60.60_{\pm1.16}$ | $42.19_{\pm12.61}$ | +0.14 | +2.80 |
| GNNML3+SIA | $92.70_{\pm0.81}$ | $70.43_{\pm6.36}$ | $64.57_{\pm2.72}$ | $37.73_{\pm7.68}$ | $69.32_{\pm3.79}$ | $48.94_{\pm10.62}$ | $63.91_{\pm5.81}$ | $48.85_{\pm12.11}$ | $61.58_{\pm3.98}$ | $49.70_{\pm17.85}$ | +0.61 | +6.45 |
| SMP | $92.30_{\pm2.62}$ | $80.25_{\pm5.98}$ | $61.32_{\pm5.69}$ | $28.71_{\pm6.55}$ | $76.87_{\pm1.90}$ | $45.69_{\pm15.96}$ | $51.09_{\pm6.39}$ | $22.98_{\pm16.26}$ | $49.15_{\pm6.92}$ | $30.73_{\pm11.30}$ | - | - |

model robust to the shift. Bevilacqua et al. (4) used a causal model to learn approximately invariant representations that better extrapolate between train and test data. Chen et al. (7) utilized structural causal models for robust out-of-distribution generalization in graph data through invariant subgraph identification and label prediction. Chu et al. (9) proposed a Wasserstein barycenter matching (WBM) layer to address the slow uncontrollable convergence rate w.r.t. graph size. Zhou et al. (41) studied the size OOD problem in the task of link prediction. Ji et al. (16) curated OOD datasets for AI-aided drug discovery. Our study stands out as the first to utilize spectral analysis to characterize the types of size-induced distribution shifts, shedding light on the underlying causes that hinder GNNs from effectively generalizing to large graphs. Some expressive models also exhibit robustness in size generalization. Murphy et al. (24) proposed an expressive model-agnostic framework that learns graph representations invariant to graph isomorphism given variable-length inputs. Clement et al. (30) proposed an expressive graph neural network that performs well on difficult structural tasks, such as cycle detection and diameter computation. Our study validates that expressive models excelling in cycle-related tasks demonstrate good size generalizability.

## 7 CONCLUSION

In conclusion, our work extensively characterizes size-induced distribution shifts and evaluates their impact on GNNs' generalizability to significantly larger test graphs compared to the training set. Spectral analysis on real-world biological data reveals a strong correlation between graph spectrum and size, which hinders GNNs' size generalization. We identify the pivotal role of cycle-related information in reducing spectral differences between small and large graphs. Motivated by these findings, we introduce three model-agnostic strategies—self-supervision, augmentation, and size-insensitive attention—to enhance GNNs' size generalizability. Empirical results show that all three strategies improve GNNs' size generalizability, with `+SIA` being the most effective. This research provides valuable insights for enhancing GNN generalization across varying graph sizes.

