## A  PROPOSITION

In this Appendix, we provide theoretical reasoning for the proposition in Section 3.1.

Without loss of generality, the output of a spectral GNN at the $(l+1)$-th layer is given by: $\mathbf{X}^{(l+1)} = \sigma(\mathbf{U}f(\mathbf{\Lambda})\mathbf{U}^T\mathbf{X}^{(l)}\mathbf{W}^{(l)})$ (Section 2). To ease our analysis, we rewrite $\mathbf{U}$, $\mathbf{\Lambda}$ and $\mathbf{X}^{(l)}$ as: $\mathbf{U} = [\mathbf{U}_1, \mathbf{U}_2, \cdots, \mathbf{U}_N]$, $\mathbf{\Lambda} = \texttt{Diag}([\lambda_1, \lambda_2, \cdots, \lambda_N])$, and $\mathbf{X}^{(l)} = [\mathbf{X}_1^{(l)}, \mathbf{X}_2^{(l)}, \cdots, \mathbf{X}_D^{(l)}]$, respectively. $\mathbf{U}_i$ is the $i$-th column vector of $\mathbf{U}$, $\lambda_i$ is the $i$-th largest eigenvalue, $\texttt{Diag}(\cdot)$ creates a matrix with the vector as its diagonal elements, $\mathbf{X}_i^{(l)}$ is the $i$-th column vector of $\mathbf{X}^{(l)}$, and $D$ is the feature dimension at the $l$-th layer.

$$\begin{aligned}
\mathbf{X}^{(l+1)} = &\sigma([\mathbf{U}_1, \mathbf{U}_2, \cdots, \mathbf{U}_N] \cdot \texttt{Diag}([f(\lambda_1), f(\lambda_2), \cdots, f(\lambda_N)]) \cdot ([\mathbf{U}_1, \mathbf{U}_2, \cdots, \mathbf{U}_N])^T \\
&\cdot [\mathbf{X}_1^{(l)}, \mathbf{X}_2^{(l)}, \cdots, \mathbf{X}_D^{(l)}] \cdot \mathbf{W}^{(l)}) \\
= &\sigma([f(\lambda_1)\mathbf{U}_1, f(\lambda_2)\mathbf{U}_2, \cdots, f(\lambda_N)\mathbf{U}_N] \cdot ([\mathbf{U}_1, \mathbf{U}_2, \cdots, \mathbf{U}_N])^T \\
&\cdot [\mathbf{X}_1^{(l)}, \mathbf{X}_2^{(l)}, \cdots, \mathbf{X}_D^{(l)}] \cdot \mathbf{W}^{(l)}).
\end{aligned} \tag{7}$$

Since $\{\mathbf{U}_i : i = 1, \cdots, N\}$ are the orthonormal basis of $\mathbb{R}^N$, $\mathbf{X}_i^{(l)}$ can be expressed as a linear combination of $\{\mathbf{U}_i\}$. Thus, suppose $\mathbf{X}_i^{(l)} = \sum_{j=1,2,\cdots,N} \alpha_j^i \mathbf{U}_j$, then Equation 7 can be rewritten as:

$$\begin{aligned}
\mathbf{X}^{(l+1)} = &\sigma([f(\lambda_1)\mathbf{U}_1, f(\lambda_2)\mathbf{U}_2, \cdots, f(\lambda_N)\mathbf{U}_N] \cdot ([\mathbf{U}_1, \mathbf{U}_2, \cdots, \mathbf{U}_N])^T \\
&\cdot [\sum_{j=1,2,\cdots,N} \alpha_j^1 \mathbf{U}_j, \cdots, \sum_{j=1,2,\cdots,N} \alpha_j^i \mathbf{U}_j, \cdots] \cdot \mathbf{W}^{(l)}) \\
= &\sigma\left([f(\lambda_1)\mathbf{U}_1, f(\lambda_2)\mathbf{U}_2, \cdots, f(\lambda_N)\mathbf{U}_N] \cdot \begin{bmatrix} \alpha_1^1 & \alpha_1^2 & \cdots & \alpha_1^D \\ \alpha_2^1 & \alpha_2^2 & \cdots & \alpha_2^D \\ \cdots & \cdots & \cdots & \cdots \\ \alpha_N^1 & \alpha_N^2 & \cdots & \alpha_N^D \end{bmatrix} \cdot \mathbf{W}^{(l)}\right) \\
= &\sigma([\sum_{j=1,\cdots,N} f(\lambda_j)\alpha_j^1 \mathbf{U}_j, \sum_{j=1,\cdots,N} f(\lambda_j)\alpha_j^2 \mathbf{U}_j, \cdots, \sum_{j=1,\cdots,N} f(\lambda_j)\alpha_j^D \mathbf{U}_j] \cdot \mathbf{W}^{(l)}).
\end{aligned} \tag{8}$$

We know that $\alpha_j^i = \mathbf{U}_j^T \cdot \mathbf{X}_i^{(l)} = \texttt{COSINE}(\mathbf{U}_j, \mathbf{X}_i^{(l)}) \cdot \|\mathbf{X}_i^{(l)}\|_2 = \texttt{COSINE}(\mathbf{U}_j, \mathbf{X}_i^{(l)}) \cdot \frac{\|\mathbf{X}_i^{(l)}\|_2}{\sqrt{N}} \cdot \sqrt{N}$. Then Equation 8 can be rewritten as:

$$\begin{aligned}
\mathbf{X}^{(l+1)} = \sigma([&\sum_{j=1,\cdots,N} f(\lambda_j)\texttt{COSINE}(\mathbf{U}_j, \mathbf{X}_1^{(l)}) \cdot \frac{\|\mathbf{X}_1^{(l)}\|_2}{\sqrt{N}} \cdot (\sqrt{N} \cdot \mathbf{U}_j), \\
&\cdots, \sum_{j=1,\cdots,N} f(\lambda_j)\texttt{COSINE}(\mathbf{U}_j, \mathbf{X}_D^{(l)}) \cdot \frac{\|\mathbf{X}_D^{(l)}\|_2}{\sqrt{N}} \cdot (\sqrt{N} \cdot \mathbf{U}_j)] \cdot \mathbf{W}^{(l)}).
\end{aligned} \tag{9}$$

The final graph representation is obtained through a global pooling (Section 2), where a set function is applied to each feature dimension. To ensure that the graph representation is irrelevant to the graph size, it is crucial for each column of $\mathbf{X}^{(l+1)}$ to be unaffected by the size. If the distributions of eigenvalues ($\lambda_j$), scaled eigenvectors ($\sqrt{N} \cdot \mathbf{U}_j$), and the scalar $\frac{\|\mathbf{X}_i^{(l)}\|_2}{\sqrt{N}}$, are uncorrelated with the graph size, then the graph representation will be size-invariant. It is worth noting that $\mathbf{X}_i^{(l)}$ contains $N$ elements and its 2-norm scales with $\sqrt{N}$; and $\|\mathbf{U}_j\|_2 = 1$ and its elements diminish with $\sqrt{N}$. Hence, we account for the scaling factor $\sqrt{N}$ in both terms. However, the distributions of eigenvalues (Section 3.2) and scaled eigenvectors in real biological graphs are influenced by the graph size. This poses a challenge for the size generalization of GNNs. Importantly, the difference in eigenvalue distributions cannot be attributed to the value range (3), as the eigenvalues of a normalized Laplacian matrix and adjacency matrix are constrained within the intervals [0,2] and [-1,1] respectively.

# B DATASETS

In this section, we provide additional details about the datasets utilized in our study. We utilize five pre-processed biological datasets, namely BBBP and BACE from the Open Graph Benchmark ([14]), and PROTEINS, NCI1, and NCI109 from TuDataset ([23]), for graph classification in our experiments. Each graph in BBBP and BACE represents a molecule, where nodes are atoms, and edges are chemical bonds. Each node has a 9-dimensional feature vector, which contains its atomic number and chirality, as well as other additional atom features such as formal charge and whether the atom is in the ring or not ([14]). In the PROTEINS dataset, nodes correspond to amino acids, and an edge connects two nodes if their distance is less than 6 Angstroms. Each node is associated with a 3-dimensional feature vector representing the type of secondary structure elements, such as helix, sheet, or turn. Each graph in NCI1 and NCI109 represents a chemical compound: each node stands for an atom and has a one-hot encoded feature representing the corresponding atom type; each edge represents the chemical bonds between the atoms. The description of each dataset is summarized as follows:

- **BBBP**: The blood-brain barrier penetration (BBBP) dataset comes from a study on the modeling and the prediction of barrier permeability. It includes binary labels of over 2000 compounds on their permeability properties ([33]).

- **BACE**: The BACE dataset provides quantitative (IC50) and qualitative (binary label) binding results for a set of inhibitors of human $\beta$-secretase 1 (BACE-1). It contains 1522 compounds with their 2D structures and binary labels ([33]).

- **PROTEINS**: The PROTEINS dataset comprises the macromolecule graphs of proteins and binary labels of the protein function (being an enzyme or not) for a total of 1113 samples ([23]).

- **NCI1 & NCI109**: The NCI1 and NCI109 are two balanced subsets of chemical compounds screened for their activity against non-small cell lung cancer and ovarian cancer cell lines, respectively ([32]). NCI1 contains a total of 4110 samples and NCI109 contains a total of 4117 samples.

# C DATA SPLITS AND PRE-PROCESSING

Table 5: Label distributions of BBBP, BACE, PROTEINS, NCI1 and NCI109 in the entire dataset, the smallest 50% subset, and the largest 10% subset.

| Dataset | | BBBP | BACE | PROTEINS | NCI1 | NCI109 |
|---|---|---|---|---|---|---|
| Entire Dataset | Number of Class 0 Samples | 479 | 822 | 663 | 2053 | 2048 |
| | Number of Class 1 Samples | 1560 | 691 | 450 | 2057 | 2079 |
| Smallest 50% | Number of Class 0 Samples | 138 | 501 | 232 | 1283 | 1283 |
| | Number of Class 1 Samples | 882 | 256 | 325 | 772 | 781 |
| Largest 10% | Number of Class 0 Samples | 122 | 53 | 101 | 78 | 87 |
| | Number of Class 1 Samples | 82 | 99 | 11 | 333 | 326 |

In this section, we discuss the techniques employed to address the challenges of class imbalance and label distribution shifts between small and large graphs. Given our emphasis on size shift, we preprocess the dataset to mitigate the influence from other distribution shifts.

For each dataset, Table 5 presents the respective class size in the whole dataset, the smallest 50% subset, and the largest 10% subset. This table reveals two main issues. Firstly, many datasets (BBBP, BACE, and PROTEINS) demonstrate imbalanced label distributions. Secondly, both the smallest 50% and the largest 10% graphs exhibit class imbalance, and there is a notable difference in the label distribution between these subsets. To mitigate these issues, we carefully split our data and apply the upsampling technique.

**Data splits.** For each dataset, we create four distinct splits: train, validation, small_test, and large_test. The large_test split consists of graphs with significantly larger sizes compared to the other splits. To

generate the train, validation, and small_test subsets, we initially select the smallest 50% of graphs from the dataset. The division among these subsets follows a ratio of 0.7:0.15:0.15, respectively. Importantly, we ensure that the data is split for each class using the same ratio, thereby maintaining consistent label distributions across the train, validation, and small_test subsets. Next, we generate the large_test subset by selecting the same number of graphs per class as in the small_test subset. The selection process begins with the largest graph within each class and ensures that the large_test subset maintains the same class distribution as the small_test subset. Table 6 shows the class size obtained after applying this operation.

**Upsampling.** As can be seen in Table 5, even after performing the appropriate data split, each dataset still exhibits varying degrees of class imbalance. To avoid training an extremely biased model, we adopt the upsampling technique during the training process. Specifically, we upsample the graphs in class 0 of BBBP at a ratio of 6, the graphs in class 1 of BACE at a ratio of 2, $\frac{2}{3}$ of the graphs in class 0 of PROTEINS at a ratio of 2, and $\frac{2}{3}$ of the graphs in class 1 of NCI1 and NCI109 at a ratio of 2.

Table 6: Label distributions after proper data splits. both the Both the small_test and large_test subsets now have the same class distribution.

|  | Dataset | BBBP | BACE | PROTEINS | NCI1 | NCI109 |
|---|---|---|---|---|---|---|
| Train | Number of Class 0 Samples | 96 | 350 | 140 | 898 | 898 |
|  | Number of Class 1 Samples | 617 | 179 | 210 | 540 | 546 |
| Val | Number of Class 0 Samples | 22 | 76 | 30 | 193 | 193 |
|  | Number of Class 1 Samples | 133 | 39 | 46 | 117 | 118 |
| Small_test | Number of Class 0 Samples | 20 | 75 | 30 | 192 | 192 |
|  | Number of Class 1 Samples | 132 | 38 | 45 | 115 | 117 |
| Large_test | Number of Class 0 Samples | 20 | 75 | 30 | 192 | 192 |
|  | Number of Class 1 Samples | 132 | 38 | 45 | 115 | 117 |

# D   DETAILS AND DISCUSSIONS ON BREAKING CYCLES AND ALIGNING CYCLE LENGTHS

In this section, we begin by elaborating on the procedures for breaking cycles and aligning cycle lengths, which we briefly introduced in Section 3. Then, we present the results of an ablation study on adding nodes randomly (instead of based on the underlying cycles).

## D.1   IMPLEMENTATION DETAILS

Our **cycle-breaking algorithm** is designed to remove all cycles in the graph with minimal edge removal while maintaining the current number of disconnected components. This approach minimally perturbs the graph while focusing on cycles. To achieve this, we begin by obtaining the cycle basis $\mathcal{C}$ (Section 2) of the input graph $\mathcal{G}$. Breaking all cycles can be accomplished by removing just one edge from each cycle in the basis. To ensure that this edge removal process does not increase the number of disconnected components, we propose a backtracking algorithm ( Algorithm 1). We note that while it is possible for Algorithm 1 to have failure cases, luckily, in all the datasets we have examined, we have not observed any instances where cycle breaking fails (i.e., Algorithm 1 returning False).

Before delving into the cycle length alignment algorithm, we first present the average statistics for both the small and large graphs, as outlined in Table 7. To derive these statistics, we obtain the cycle basis (Section 2) for each input graph and compute the average length across the cycles in that basis. Then, we calculate the average cycle length by further averaging it over all the graphs within the small and large datasets, respectively. Note that small and large graphs differ in both the mean and the standard deviation of cycle lengths, indicating that large graphs may contain notably long cycles that are not present in the small graphs.

---

**Algorithm 1** Cycle Breaking

---

1: **procedure** CYCLEBACKTRACKING($i$: index integer, $\mathcal{G}$: input graph, $\mathcal{C}$: a list of cycle basis)
2:     **if** $i == \text{len}(\mathcal{C})$ **then**
3:         **return** True
4:     $N_{\text{before}} \leftarrow \text{NumberOfConnectedComponent}(\mathcal{G})$
5:     **for** edge pair $\varepsilon$ in $\mathcal{C}_i$ **do**
6:         $\mathcal{G} \leftarrow \text{Remove}(\varepsilon)$
7:         $N_{\text{after}} \leftarrow \text{NumberOfConnectedComponent}(\mathcal{G})$
8:         **if** $N_{\text{before}} == N_{\text{after}}$ **then**
9:             **if** CycleBacktracking$(i + 1, \mathcal{G}, \mathcal{C})$ **then**
10:                 **return** True
11:         $\mathcal{G} \leftarrow \text{add}(\varepsilon)$
12:     **return** False

---

Table 7: Average cycle lengths for small and large graphs before and after cycle length alignment. Small and large graphs in real biological datasets originally differ in cycle length and such difference is reduced after cycle length alignment.

| | BBBP | | BACE | | NCI1 | | NCI109 | | PROTEINS | |
|---|---|---|---|---|---|---|---|---|---|---|
| | small | large | small | large | small | large | small | large | small | large |
| before alignment | $5.56_{\pm 1.39}$ | $6.54_{\pm 3.52}$ | $5.70_{\pm 0.56}$ | $6.50_{\pm 2.01}$ | $5.64_{\pm 1.19}$ | $7.04_{\pm 3.05}$ | $5.59_{\pm 1.25}$ | $7.06_{\pm 2.88}$ | $3.20_{\pm 0.34}$ | $6.24_{\pm 2.75}$ |
| after alignment | $6.50_{\pm 3.12}$ | $6.54_{\pm 3.52}$ | $6.48_{\pm 2.08}$ | $6.50_{\pm 2.01}$ | $6.97_{\pm 2.97}$ | $7.04_{\pm 3.05}$ | $6.86_{\pm 3.02}$ | $7.06_{\pm 2.88}$ | $5.97_{\pm 3.03}$ | $6.24_{\pm 2.75}$ |

Our **cycle length alignment algorithm** aligns both the *mean* and *std* of average cycle lengths between small and large graphs. This is achieved by selectively increasing the cycle length of a subset of small graphs. Formally, we present our algorithm in Algorithm 3. It has two hyperparameters: $n$, which controls the maximum cycle length increments one graph can get, and $\mathcal{R}$, which controls the portion of small graphs whose cycle lengths will be increased. We tune these hyperparameters in order to best align the cycle length of small graphs with that of large graphs. We tune $n$ over $[2, 3, 4, 5, 6, 7]$ and $\mathcal{R}$ over $[1, 2, 3, 4, 5, 6, 7, 8]$, and identify $n = 7, \mathcal{R} = 8$ for BBBP, $n = 6, \mathcal{R} = 8$ for BACE, $n = 6, \mathcal{R} = 5$ for NCI1, $n = 6, \mathcal{R} = 5$ for NCI109 and $n = 5, \mathcal{R} = 3$ for PROTEINS. We present the average cycle length after alignment in Table 7 and the discrepancy between small and large graphs is significantly reduced.

---

**Algorithm 2** Add One Cycle Length

---

1: **procedure** ADD1_CYCLE_LENGTH($\mathcal{G}$: input graph, $\mathcal{C}$: list of cycle basis)
2:     **for** each cycle $\mathcal{C}_i$ **do**
3:         edge $(v_1, v_2) \leftarrow \text{RandomChoice}(\mathcal{C}_i)$
4:         $\mathcal{G} \leftarrow \text{RemoveEdge}(v_1, v_2)$
5:         $v_{\text{new}} \leftarrow \text{ReplicateNode}(\arg\min_{\text{degree}}(\mathcal{C}_i))$
6:         $\mathcal{G} \leftarrow \text{AddNode}(v_{\text{new}})$
7:         $\mathcal{G} \leftarrow \text{AddEdge}(v_1, v_{\text{new}}) \text{ AddEdge}(v_2, v_{\text{new}})$
8:     **return** $\mathcal{G}$

---

### D.2 ABLATION STUDY ON RANDOMLY ADDING NODES

One natural question arising from Section 3.3.1 is whether the cycle length alignment is necessary or it suffices to add nodes / edges randomly in order to reduce the spectrum discrepancy between small and large graphs. To answer this question, we conduct an ablation study on adding nodes and edges randomly to the graphs in the small dataset, *matching the quantity* introduced during the cycle length alignment process. We then compare the changes in the relative spectrum difference with those observed during the cycle length alignment procedure.

We present our results in Table 8. We find that aligning cycle lengths consistently leads to smaller spectrum differences between graphs of varying sizes compared to randomly adding nodes. These results further highlight the importance of cycle lengths for GNNs to achieve size generalizability.

---

**Algorithm 3** Align Cycle Length

---

1: **procedure**  ALIGNCYCLELENGTH($\{\mathcal{G}_i\}_{i=1}^N$: a set of graphs,  $\mathcal{R}$: skipping ratio, $n$: increased cycle length, $n \geq 1$)
2:  ResultList $\mathcal{M} \leftarrow []$
3:  **for** $i = 0, 1, \ldots, N - 1$ **do**
4:    $\mathcal{G}' \leftarrow \mathcal{G}_i$
5:    **if** $i \mod \mathcal{R} == 0$ **then**
6:      **for** $j = 0, 1, \ldots, n - 1$ **do**
7:        obtain Cycle Basis $\mathcal{C} \leftarrow \mathcal{G}'$
8:        $\mathcal{G}' \leftarrow$ Add1_Cycle_Length($\mathcal{G}', \mathcal{C}$)
9:    $\mathcal{M} \leftarrow$ append $\mathcal{G}'$
10:  **return** $\mathcal{M}$

---

Table 8: Comparison between randomly adding nodes and aligning cycle lengths. Average Wasserstein distance of eigenvalue distributions between graphs of similar size and different sizes are computed. Relative difference is computed as in Table 1. We use ↑ (↓) to denote the increase (decrease) in the relative difference compared to not taking the corresponding action. Aligning cycle lengths results in a greater reduction of the relative spectrum difference compared to randomly adding a node.

| | Randonly adding nodes | | | Aligning cycle lengths | | |
|---|---|---|---|---|---|---|
| Datasets | Different sizes | Similar size | △ relative difference | Different sizes | Similar size | △ relative difference |
| **BBBP** | 0.00557 | 0.00203 | ↓ 33% | 0.00565 | 0.00211 | ↓ 41% |
| **BACE** | 0.00406 | 0.00162 | ↓ 26% | 0.00417 | 0.00176 | ↓ 41% |
| **NCI1** | 0.00559 | 0.00230 | ↓ 20% | 0.00566 | 0.00242 | ↓ 31% |
| **NCI109** | 0.00558 | 0.00232 | ↓ 22% | 0.00568 | 0.00245 | ↓ 31% |
| **PROTEINS** | 0.00756 | 0.00280 | ↓ 24% | 0.00763 | 0.00302 | ↓ 41% |

## E  ABLATION STUDIES ON ATTENTION MECHANISM

Table 9: Size generalizability evaluated with +NA, following the same rule as in Table 3. Naive attention is not helpful for size generalization.

| Datasets | BBBP | | BACE | | PROTEINS | | NCI1 | | NCI109 | | Avg Improv | |
|---|---|---|---|---|---|---|---|---|---|---|---|---|
| Models | Small | Large | Small | Large | Small | Large | Small | Large | Small | Large | Small | Large |
| **MLP** | $90.36_{\pm0.71}$ | $55.61_{\pm3.37}$ | $61.06_{\pm5.79}$ | $21.06_{\pm7.89}$ | $36.15_{\pm2.28}$ | $21.55_{\pm1.34}$ | $36.43_{\pm3.89}$ | $3.36_{\pm2.87}$ | $35.87_{\pm4.23}$ | $4.65_{\pm3.72}$ | - | - |
| **MLP+NA** | $88.83_{\pm1.40}$ | $62.09_{\pm10.27}$ | $56.38_{\pm15.89}$ | $21.34_{\pm8.52}$ | $38.87_{\pm13.04}$ | $29.42_{\pm23.83}$ | $28.89_{\pm5.96}$ | $1.98_{\pm0.98}$ | $39.20_{\pm2.96}$ | $1.55_{\pm0.01}$ | -1.71 | +2.03 |
| **MLP+SIA** | $90.38_{\pm1.05}$ | $62.79_{\pm7.55}$ | $60.85_{\pm7.83}$ | $21.79_{\pm15.07}$ | $40.68_{\pm3.56}$ | $33.57_{\pm11.87}$ | $35.42_{\pm3.83}$ | $3.26_{\pm2.55}$ | $39.20_{\pm2.96}$ | $12.2_{\pm5.05}$ | +1.33 | +5.48 |
| **GCN** | $91.37_{\pm0.59}$ | $68.59_{\pm7.47}$ | $63.68_{\pm6.63}$ | $28.72_{\pm14.26}$ | $72.35_{\pm2.58}$ | $40.57_{\pm7.67}$ | $54.91_{\pm2.37}$ | $28.80_{\pm7.57}$ | $60.83_{\pm1.92}$ | $30.45_{\pm4.34}$ | - | - |
| **GCN+NA** | $89.45_{\pm1.65}$ | $64.38_{\pm7.10}$ | $62.03_{\pm1.12}$ | $28.19_{\pm16.32}$ | $69.95_{\pm3.27}$ | $52.41_{\pm8.71}$ | $60.37_{\pm1.40}$ | $31.52_{\pm4.18}$ | $54.97_{\pm2.57}$ | $27.60_{\pm10.07}$ | -1.27 | -0.41 |
| **GCN+SIA** | $91.32_{\pm0.73}$ | $71.66_{\pm6.99}$ | $64.35_{\pm9.76}$ | $24.24_{\pm17.03}$ | $73.84_{\pm3.65}$ | $58.74_{\pm9.49}$ | $59.78_{\pm1.65}$ | $45.70_{\pm6.70}$ | $60.32_{\pm2.90}$ | $38.78_{\pm4.55}$ | +1.29 | +8.40 |
| **GAT** | $91.27_{\pm1.43}$ | $68.35_{\pm7.02}$ | $69.73_{\pm2.05}$ | $42.23_{\pm11.18}$ | $72.25_{\pm4.25}$ | $43.86_{\pm6.82}$ | $58.22_{\pm2.86}$ | $49.36_{\pm4.12}$ | $64.39_{\pm3.29}$ | $38.36_{\pm8.93}$ | - | - |
| **GAT+NA** | $91.16_{\pm1.45}$ | $70.08_{\pm6.42}$ | $63.42_{\pm2.64}$ | $34.08_{\pm13.43}$ | $72.25_{\pm2.24}$ | $53.25_{\pm9.07}$ | $60.98_{\pm0.93}$ | $29.76_{\pm2.58}$ | $60.18_{\pm5.91}$ | $35.24_{\pm5.82}$ | -1.57 | -3.95 |
| **GAT+SIA** | $91.88_{\pm2.12}$ | $74.87_{\pm5.62}$ | $69.64_{\pm6.79}$ | $43.87_{\pm7.98}$ | $75.35_{\pm3.28}$ | $62.71_{\pm4.98}$ | $61.42_{\pm1.07}$ | $55.73_{\pm12.98}$ | $63.27_{\pm3.15}$ | $45.97_{\pm7.74}$ | +1.14 | +8.20 |
| **GIN** | $88.28_{\pm2.39}$ | $66.67_{\pm5.55}$ | $57.02_{\pm6.48}$ | $22.97_{\pm10.26}$ | $74.55_{\pm4.27}$ | $50.20_{\pm5.36}$ | $62.17_{\pm3.86}$ | $44.26_{\pm7.03}$ | $62.42_{\pm2.77}$ | $33.23_{\pm6.77}$ | - | - |
| **GIN+NA** | $89.59_{\pm2.64}$ | $69.89_{\pm12.97}$ | $55.46_{\pm10.59}$ | $18.92_{\pm15.76}$ | $73.59_{\pm4.06}$ | $54.14_{\pm8.14}$ | $61.15_{\pm4.62}$ | $35.73_{\pm9.73}$ | $61.62_{\pm3.08}$ | $32.73_{\pm8.67}$ | -0.61 | -1.18 |
| **GIN+SIA** | $92.70_{\pm0.45}$ | $75.99_{\pm4.74}$ | $61.30_{\pm6.77}$ | $24.42_{\pm16.37}$ | $74.88_{\pm4.24}$ | $51.36_{\pm7.76}$ | $62.83_{\pm1.07}$ | $42.82_{\pm8.92}$ | $63.00_{\pm4.24}$ | $41.65_{\pm4.19}$ | +2.05 | +3.78 |
| **FAGCN** | $90.58_{\pm1.72}$ | $64.93_{\pm7.62}$ | $62.96_{\pm2.12}$ | $24.65_{\pm11.71}$ | $70.03_{\pm5.20}$ | $42.34_{\pm6.61}$ | $43.51_{\pm4.29}$ | $10.16_{\pm7.80}$ | $55.78_{\pm3.5}$ | $22.65_{\pm12.87}$ | - | - |
| **FAGCN+NA** | $91.97_{\pm1.67}$ | $72.30_{\pm10.54}$ | $58.39_{\pm4.25}$ | $10.71_{\pm9.0}$ | $70.47_{\pm4.73}$ | $51.61_{\pm12.47}$ | $50.27_{\pm4.45}$ | $12.11_{\pm5.39}$ | $50.75_{\pm10.91}$ | $10.26_{\pm16.24}$ | -0.20 | -1.55 |
| **FAGCN+SIA** | $90.17_{\pm2.83}$ | $74.65_{\pm9.13}$ | $62.40_{\pm3.36}$ | $30.35_{\pm13.48}$ | $71.30_{\pm5.79}$ | $48.94_{\pm10.62}$ | $46.95_{\pm5.71}$ | $10.99_{\pm7.50}$ | $52.82_{\pm6.28}$ | $19.08_{\pm5.32}$ | +0.16 | +3.86 |
| **GNNML3** | $92.01_{\pm1.56}$ | $64.18_{\pm6.99}$ | $62.31_{\pm4.90}$ | $32.94_{\pm12.86}$ | $71.59_{\pm3.5}$ | $40.74_{\pm15.0}$ | $63.73_{\pm4.67}$ | $51.75_{\pm9.05}$ | $59.39_{\pm3.76}$ | $33.80_{\pm11.19}$ | - | - |
| **GNNML3+NA** | $87.85_{\pm1.96}$ | $66.42_{\pm1.93}$ | $65.15_{\pm7.50}$ | $42.05_{\pm13.30}$ | $66.11_{\pm10.07}$ | $55.39_{\pm13.23}$ | $59.58_{\pm5.06}$ | $32.48_{\pm8.90}$ | $60.76_{\pm4.95}$ | $27.18_{\pm8.65}$ | -1.92 | +0.02 |
| **GNNML3+SIA** | $92.70_{\pm0.81}$ | $70.43_{\pm6.36}$ | $64.57_{\pm2.72}$ | $37.73_{\pm7.68}$ | $69.32_{\pm3.79}$ | $48.94_{\pm10.62}$ | $63.91_{\pm5.81}$ | $48.85_{\pm12.11}$ | $61.58_{\pm3.98}$ | $49.70_{\pm17.85}$ | +0.61 | +6.45 |

The results in Table 9 demonstrate that +NA does not consistently enhance graph classification for large graphs. This suggests that relying solely on the attention mechanism, without considering cycle information, does not improve the size generalizability of GNNs. In contrast, the incorporation of cycle information in +SIA leads to significant improvements on large graphs.

## F  TRAINING DETAILS

In this section, we provide the details of our training process for the purpose of reproducibility.

Table 10: Average Wasserstein distance between graphs of 'similar sizes' and graphs of 'different sizes' based on **degree** distributions, respectively. The relative difference is computed by the difference of the Wasserstein distance normalized by the Wasserstein distance of similar graphs.

|  | BBBP | BACE | PROTEINS | NCI109 | NCI1 |
|---|---|---|---|---|---|
| **Different Size** | 0.00177 | 0.00120 | 0.00062 | 0.00161 | 0.00163 |
| **Similar Size** | 0.00157 | 0.00105 | 0.00054 | 0.00146 | 0.00148 |
| **Relative Difference** | 12.1% | 14.6% | 14.8% | 10.1% | 9.8% |

To ensure fair comparisons, we maintain consistent hyperparameter settings across all experiments. Specifically, we use a batch size of 30 and a learning rate of 0.001. The Adam optimizer is employed without gradient clipping. For the baseline models, we utilize three graph convolution layers with a global max pooling. The previously described upsampling technique is applied to all methods. To prevent overfitting, we implement early stopping with a patience period of 50 epochs, and the lowest validation loss as the criterion for model selection. Each method is executed five times, and we report the average result along with the standard deviation in Table 3 and Table 4.

To ensure consistency with the original papers, we set the other hyperparameters of each baseline model to match the values specified in their respective papers (except for the threshold in +SAGPool). In the case of thresholded SAG pooling, we conduct tuning with thresholds from the set $\{1e-1, 2e-1, 5e-2, 1e-2, 1e-3, 1e-4, 5e-1\}$. Based the validation results, we identify a threshold of $1e-2$ for BBBP, BACE, $5e-1$ for PROTEINS, and $1e-4$ for NCI and NCI109. For +SSR, we use the same hyperparameter settings as in (6) and set the regularization coefficient $\lambda = 0.1$ and coarsening ratios $C = \{0.8, 0.9\}$ for all datasets. We adhere to the authors' recommendation and employ spectral graph coarsening as the coarsening algorithm. Additionally, we fine-tune the aggregation strategy, selecting from options such as max, mean, and sum, to derive features for the nodes in the coarsened version. For SMP and +RPGNN, we use the original code provided by the authors. For SMP, we tune the number of SMP and FastSMP layers from the set {3,4}. Based on the validation results, we implement 3 FastSMP layers. For +RPGNN, we use the same hyperparameter settings as in (4; 6). For +CIGA, we follow the same hyperparameter search range as specified in (7). Regarding the +SSL method that we proposed, we tune $\lambda$ over $\{1, 1e-1, 1e-2, 1e-3, 1e-4, 1e-5\}$. Based on the validation results, we identify $\lambda = 1e-2$ for NCI1, and NCI109, $\lambda = 1$ for PROTEINS, BBBP, and $\lambda = 1e-2$ for BACE. For our proposed cycle augmentation, we use the set of hyperparameters $n, \mathcal{R}$ from Appendix D.1. For +SIA, we have no hyperparameters to tune.

## G  ADDITIONAL PLOTS ON DEGREE DISTRIBUTIONS

Figure 3 and Table 10 reveal that the degree distribution does not exhibit a clear correlation with the graph size. These visualizations and tables adhere to the same convention as presented in Section 3.2. Unlike the assumption made in (38) that attributes changes in local degree patterns as the primary covariate shifts induced by size, our investigation on biological datasets suggests a different perspective.

## H  ADDITIONAL RESULTS ON REAL-WORLD DATASETS

In this section, we demonstrate +SIA's effectiveness in size generalization using the HIV dataset from (15). Adopting the ROC-AUC metric from (7; 15), we observe that +SIA still consistently achieves performance improvements across all backbone models.

Table 11: Size generalizability evaluated on the real-world hiv datasets. +SIA still leads to consistent performance boosts.

| Dataset | Hiv | | | | | | | | | | | |
|---|---|---|---|---|---|---|---|---|---|---|---|---|
| Models | MLP | | GCN | | GAT | | GIN | | FAGCN | | GNNML3 | |
|  | Small | Large | Small | Large | Small | Large | Small | Large | Small | Large | Small | Large |
| Original Performance | $67.95_{\pm2.22}$ | $38.97_{\pm6.39}$ | $71.29_{\pm1.58}$ | $38.35_{\pm4.09}$ | $73.34_{\pm1.77}$ | $41.23_{\pm4.62}$ | $71.04_{\pm1.45}$ | $43.65_{\pm3.01}$ | $68.35_{\pm3.09}$ | $50.03_{\pm1.85}$ | $74.32_{\pm1.13}$ | $47.44_{\pm7.66}$ |
| +SIA Performance | $70.64_{\pm1.94}$ | $40.93_{\pm2.87}$ | $70.27_{\pm1.42}$ | $43.24_{\pm6.08}$ | $72.33_{\pm1.72}$ | $46.72_{\pm2.74}$ | $69.92_{\pm1.50}$ | $43.55_{\pm3.84}$ | $69.53_{\pm1.92}$ | $52.35_{\pm6.35}$ | $74.52_{\pm2.95}$ | $54.68_{\pm6.60}$ |
| +Avg Improvements | (small) +0.15 ; (large) +3.63 | | | | | | | | | | | |

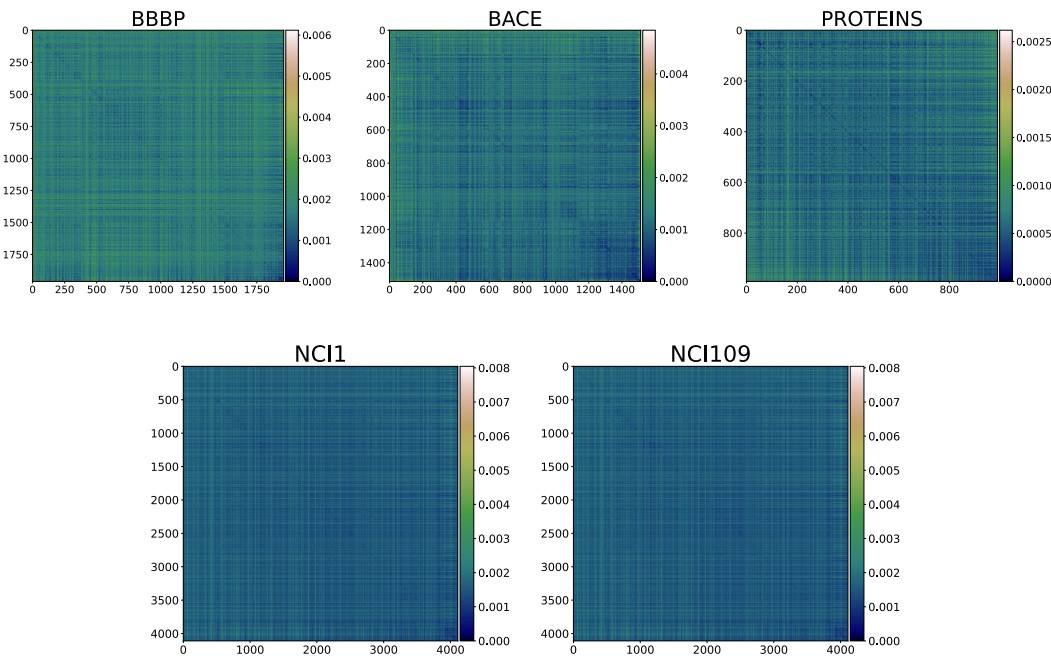

Figure 3: Pairwise graph distance measured by the Wasserstein distance of degree distributions, following the same convention in Figure 1. **Degree distribution does not show a clear correlation with graph size**.

# I  ADDITIONAL PLOTS ON CYCLE BREAKING & CYCLE LENGTH ALIGNMENT

Following the same convention in Figure 2, Figure 4 presents the pairwise graph distance measured by eigenvalue distributions after breaking cycles and aligning cycles on other datasets.

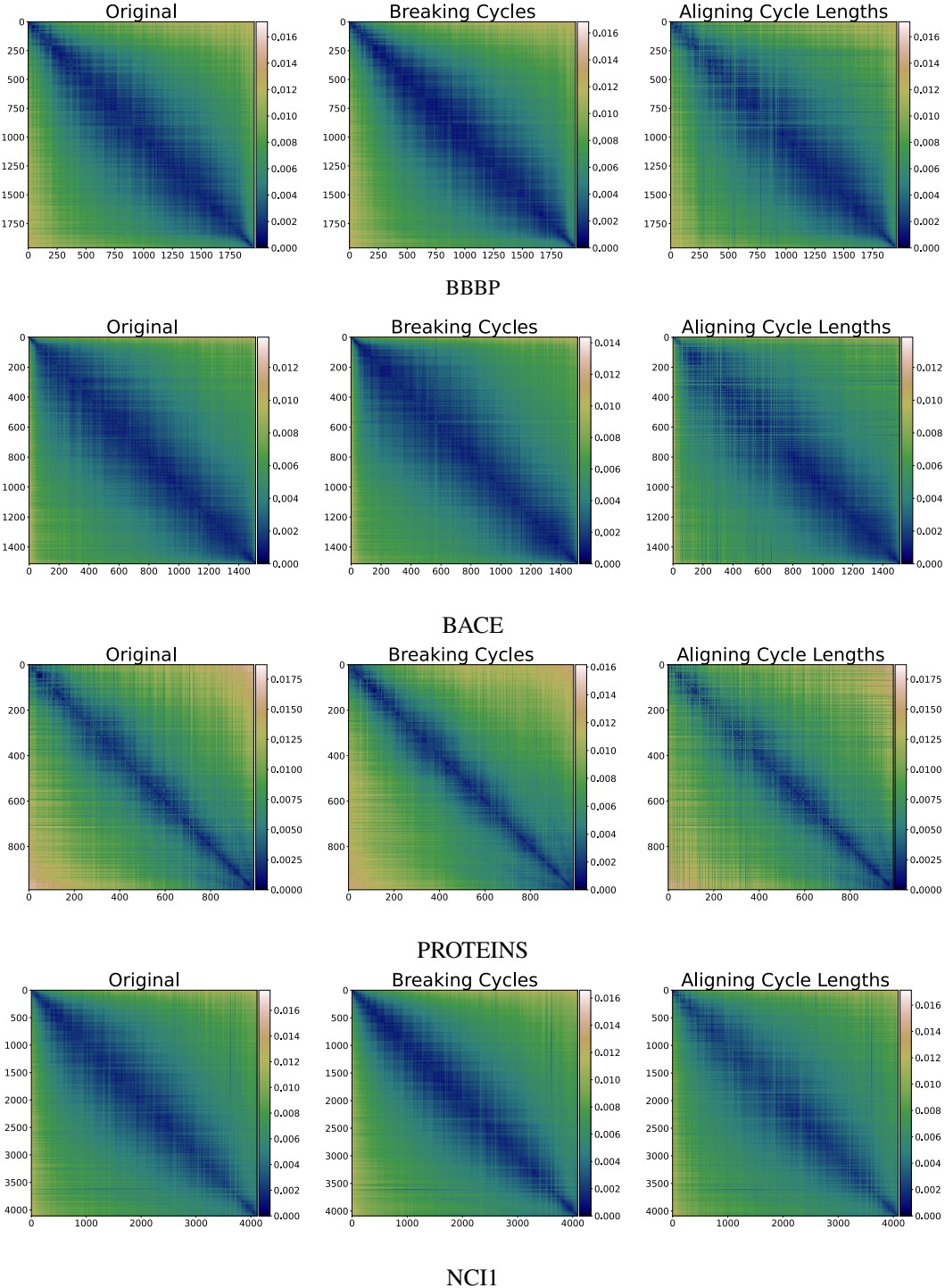

Figure 4: Pairwise graph distance measured by the Wasserstein distance of eigenvalue distributions after breaking cycles and aligning cycle lengths on different datasets. Breaking cycles **amplifies the correlation** between eigenvalue distribution and graph size, while aligning cycle lengths **reduces the correlation**.