# OpenReview forum: "Size Generalization of Graph Neural Networks on Biological Data: Insights and Practices from the Spectral Perspective"
_ICLR.cc/2024/Conference — Submitted to ICLR 2024_

### Official Review · Reviewer_Xenz · 2023-10-27

**Soundness:** 3 good
**Presentation:** 3 good
**Contribution:** 2 fair
**Rating:** 6
**Confidence:** 3

**Summary:**

The paper investigates how graph neural networks (GNNs) handle graphs of different sizes, particularly focusing on their ability to generalize from smaller to larger graphs. Using biological datasets, the authors adopt a spectral analysis approach to show that differences in subgraph patterns, like cycle lengths, affect a GNN's performance when it encounters larger graphs. They propose and compare three model-agnostic strategies—self-supervision, augmentation, and size-insensitive attention—to enhance GNNs' size generalizability, finding that size-insensitive attention is the most effective method for improving performance on larger graphs.

**Strengths:**

* The paper starts with the study of types of distribution shifts happening real-world graphs and provides several insights, in particular to cycle importance.
* The paper proposes and compares 3 different model-agnostic methods to enhance their performance in classification tasks.
* The experiments on classification indicates that these methods are usually universally good across different models and datasets.

**Weaknesses:**

* As the paper takes a data-driven approach, the main question is whether these empirical results are transferable to other domains,  other datasets, other models.
* Augmenting model representations with different statistics is not novel. It's not clear how their enhancements correlate with previous approaches.

**Questions:**

1. What is the time degradation when performing these augmentation? How much more time needed to perform classification?
2. The models and datasets are academic. Is it possible to apply this model to more real-world datasets and showcase how this method can be used in biological scenarios?

---

> ### Author Response · Authors · 2023-11-22
> **Response to Reviewer Xenz**
>
> We thank the reviewer for acknowledging the insights of our paper and the effectiveness of our proposed methods. We also thank the reviewer for appreciating the soundness and presentation of our work. We appreciate all the questions and feedback the reviewer raised. In the following, we carefully respond to each of the concerns and suggestions that the reviewer raised.
>
> **Q4.1 Transferability of results**
>
> As mentioned in the general response, one important insight brought up by this paper is that the covariate shifts caused by variations in sizes are dependent on the data domain. Thus, we restrict our attention to the biological domain. More importantly, most existing datasets for graph classification belong to the biological or chemistry domain.
>
> **Q4.2 Augmentation similar to prior approaches**
>
> We would like to acknowledge that augmentation is not a unique technique proposed by us and neither is that the main contribution of this paper. Instead, it provides tools to validate the main insights of this paper --- leveraging cycle information in GNN can improve size generalizability on biological data. Our insights are orthogonal to different methodology-driven papers and can be integrated with those methods.
>
> **Q4.3 Additional time for augmentation**
>
> In practice, our proposed strategies do not substantially increase training time. For cycle augmentation, we perform augmentations before training and thus do not incur overhead time during classification. For SIA, it incurs additional time due to the attention mechanism on each graph. We present per-epoch runtime statistics as below:
>
>
> |                               | bbbp | bace | proteins | NCI1 | NCI109 |
> | ----------------------------- | ---- | ---- | -------- | ---- | ------ |
> | Original (per-epoch time/sec) | 0.33 | 0.21 | 0.13     | 0.55 | 0.53   |
> | SSL (per-epoch time/sec)      | 0.33 | 0.21 | 0.12     | 0.57 | 0.54   |
> | CycleAug (per-epoch time/sec) | 0.34 | 0.21 | 0.13     | 0.54 | 0.53   |
> | SIA (per-epoch time/sec)      | 1.32 | 0.78 | 0.55     | 1.98 | 1.94   |
>
>
> **Q4.4 Other datasets**
>
> As mentioned in the general response, our selection of datasets is based on the biological domain, and the dataset should not suffer significantly from other issues that blur the focus on size generalization. More details can be found in the datasets section in the general response.

---

### Official Review · Reviewer_dD3c · 2023-10-31

**Soundness:** 2 fair
**Presentation:** 3 good
**Contribution:** 3 good
**Rating:** 5
**Confidence:** 4

**Summary:**

This paper tackles distribution shifts caused by the graph sizes of training and test sets.
First, through analysis of the spectrum distribution, it was shown that there is a correlation between the size of the graph and the distribution. It was empirically shown that the degree of correlation changes by adjusting the size of the cycle.
By this empirical evidence, this paper proposes three approaches to make GNNs aware of the existence and number of cycles.
Experimental results demonstrate the potential of graph neural networks (GNNs) to enhance size generalization by understanding their substructure.

**Strengths:**

1. Spectral analysis of size generalizability of GNNs is novel.
2. The proposed approaches to alleviate distribution shift are effective.

**Weaknesses:**

1. Lack of mathematical proof of the relationship between spectrum changes depending on the size and number of cycles.
2. GNNs that counts or can aware substructures were not compared.
3. Inappropriate experimental settings.

**Questions:**

1. The paper focuses on the relationship between cycle size and size generalizability. Could size generalizability be related to the number of cycles?

2. Where is the theoretical evidence that reveals the relation between the size/number of cycles and spectrum distribution?

3. Besides cycles, can there be any substructure that changes the spectrum according to changes in size and number?

4. Where is a comparison with GNNs [1-8] that can understand the structure of the substructure and predict its number relatively accurately or consider out-of-distribution?

&nbsp;&nbsp;&nbsp;&nbsp;&nbsp; [1] "From stars to subgraphs: Uplifting any GNN with local structure awareness." ICLR 2022.

&nbsp;&nbsp;&nbsp;&nbsp;&nbsp; [2] "Building powerful and equivariant graph neural networks with structural message-passing" NeurIPS 2020.

&nbsp;&nbsp;&nbsp;&nbsp;&nbsp; [3] "Understanding and extending subgraph gnns by rethinking their symmetries", NeurIPS 2022.

&nbsp;&nbsp;&nbsp;&nbsp;&nbsp; [4] "Nested graph neural networks", NeurIPS 2021.

&nbsp;&nbsp;&nbsp;&nbsp;&nbsp; [5] "From local structures to size generalization in graph neural networks", ICML 2021.

&nbsp;&nbsp;&nbsp;&nbsp;&nbsp; [6] "Relational pooling for graph representations", ICML 2019.

&nbsp;&nbsp;&nbsp;&nbsp;&nbsp; [7] "Size-invariant graph representations for graph classification extrapolations", ICML 2021.

&nbsp;&nbsp;&nbsp;&nbsp;&nbsp; [8] "Improving graph neural network expressivity via subgraph isomorphism counting", IEEE TPAMI 2022.

5. The results in Tables 3 and 4 are the results after class imbalance and size imbalance have been corrected. What is the performance in the class imbalance setting of the original data?

6. Is size generalizability using cycle applicable to other data domains beyond the biological domain?

**Details Of Ethics Concerns:**

None.

---

> ### Author Response · Authors · 2023-11-22
> **Response to Reviewer dD3c**
>
> We thank the reviewer for acknowledging the novelty of the spectral analysis and the effectiveness of our proposed methods. We also thank the reviewer for carefully reading our paper and detailed comments. We appreciate all the questions and feedback the reviewer raised, especially the one on drawing our attention to expressive models and the number of related works provided. In the following, we carefully respond to each of the concerns and suggestions that the reviewer raised.
>
> **Q3.1 Size generalizability and the number of cycles**
>
> Thank you for your great suggestion. We present the following table to show the statistics of the average number of cycles that small and large graphs contain respective to their size.
>
> |          | num_cycles/num_nodes for smallest 50% graphs | num_cycles/num_nodes for largest 10% graphs |
> | -------- | -------------------------------------- | -------------------------------------- |
> | bbbp     | 0.1211 +- 0.05                         | 0.1103 +- 0.03                         |
> | bace     | 0.1335 +- 0.02                         | 0.0814 +- 0.02                         |
> | PROTEINS | 0.9237 +- 0.19                         | 0.8320 +- 0.26                         |
> | NCI1     |  0.1276 +- 0.05                         | 0.1103 +- 0.05                         |
> | NCI109   | 0.1282 +- 0.05                         | 0.1127 +- 0.05                         |
>
> It can be seen that the average num_cycle/num_nodes are highly similar for small and large graphs for most datasets. Due to this reason, it is unlikely that the number of cycles is closely related to size generalization.
>
> **Q3.2 Besides cycles, can there be any substructure that changes the spectrum according to changes in size and number?**
>
> There can be. The insights for our investigation into cycles are based on our discernment of variations in the peaks of specific frequencies within the graph spectrum, such as the frequency=1 in unnormalized Laplacian matrices. Many of these specific frequencies align with the frequencies of cycles, with cycles containing 5-7 nodes being particularly evident. As a reminder, the spectrum for a cycle with n nodes is given by the formula 2-2cos(2πj/n). We also computed the spectrum of other substructures such as lines, but their difference in small and large graphs is not consistent across different biological datasets.
>
> **Q3.4 Comparison with other baselines**
>
> We have added more baselines in Table 4 (including some expressive GNN models). For more details, please refer to the baseline section in the general response and Table 4 in our updated submission. In general, the proposed method SIA still achieves best performance and is model-agnostic. Thank you for your valuable suggestions, we do find that expressive model that excels at detecting cycles have good size generalizability on certain datasets.
>
> **Q3.5 performance in the class imbalance setting of the original data**
>
> In our experiments, we find that certain backbone models tend to be very biased towards the major class and thus result in an extremely skewed performance. For instance, the FAGCN and MLP backbone model always outputs a test accuracy (and thus unreliable F1 score) that is almost the same as the proportion of the larger class. In order to prevent such cases from happening, we decided to adopt upsampling.
>
> **Q3.6 Is size generalizability using cycle applicable to other data domains beyond the biological domain?**
>
> It might be also applicable to the chemistry domain. For the other domains, it relies on further investigation of the spectrums. However, most existing datasets for graph classification belong to the biological or chemistry domain.

---

### Official Review · Reviewer_XnPU · 2023-10-31

**Soundness:** 2 fair
**Presentation:** 3 good
**Contribution:** 2 fair
**Rating:** 3
**Confidence:** 5

**Summary:**

This paper characterizes the size-induced distribution shifts and evaluated their influence on the generalizability of GNNs through the spectral perspective especially on biological data. It identifies that spectrum differences induced by size are related to differences in subgraph patterns and introduces three model-agnostic strategies to enhance GNNs’ size generalizability.

**Strengths:**

This paper identifies that cycle-related information plays a pivotal role in reducing spectral differences between small and large graphs. It proposes three model-agnostic strategies—self-supervision, augmentation, and size-insensitive attention—to enhance GNNs’ size generalizability and empirical results demonstrated that their effectiveness.

**Weaknesses:**

1. Experiments are insufficient and lack of comparison with related methods, such as the size-generalization methods referenced in the related work. The baselines are not state-of-the-art methods in the relevant field. The authors need to add comparison experiments with methods from most recent years, which are related to this paper. Furthermore, the paper lacks experimental validation from other perspectives, such as the effect of different graph size settings in the training process.

2. The contribution lacks novelty. In this paper, the authors identify cycle structures as a major factor affecting the generalization capacities of GNNs. This finding looks to be a special case of [1]. In Section 3, the authors observe that cycle structures have an impact on the spectrum differences between graphs, but it is difficult to ascertain the effect of graph size and cycle distribution on the generalization capacities of models. Also, the three proposed strategies in Section 4 lack novelty and could be combined into a single algorithm.

[1] Gilad Yehudai, Ethan Fetaya, Eli Meirom, Gal Chechik, and Haggai Maron. From local structures to size generalization in graph neural networks. In International Conference on Machine Learning, pages 11975–11986. PMLR, 2021.

**Questions:**

Why is the algorithm description incomplete in the text, such as the section 4.3? If not essential, it could be excluded as the part of the contributions.

---

> ### Author Response · Authors · 2023-11-22
> **Response to Reviewer XnPU**
>
> We thank the reviewer for acknowledging the insight of this paper in identifying cycle-related information. We value the constructive feedback that the reviewer provided. In the following, we carefully respond to each of the concerns and questions that the reviewer raised.
>
>
> **Q2.1 Insufficient experiments and comparison with baselines**
>
> We have added more baselines in Table 4. For more details, please refer to the baseline section in the general response and Table 4 in our updated submission. In general, the proposed method SIA still achieves the best performance and is model-agnostic.
>
> **Q2.2 Novelty wrt. "From local structures to size generalization in graph neural networks"**
>
> Actually, the empirical findings of our paper are contradictory to the assumptions made by the paper "From local structures to size generalization in graph neural networks". In their paper, they hold the assumption that the degree patterns change with graph sizes. However, in Appendix G of our updated paper, we find that the degree distribution does not show a clear correlation with the graph size on biological datasets. Our main contribution is that we find cycle-related information (existence of cycles, average cycle length) is critical for size generalization for biological data, which has not been proposed before.
>
> **Q2.3 Why is the algorithm description incomplete in the text, such as the section 4.3? If not essential, it could be excluded as the part of the contributions.**
>
> This is primarily due to page limits. In fact, we have explained well about the motivation as well as the main function of this algorithm at the same paragraph. We leave the details of implementation in the Appendix for further interest.

---

### Official Review · Reviewer_eDSS · 2023-11-01

**Soundness:** 3 good
**Presentation:** 3 good
**Contribution:** 3 good
**Rating:** 6
**Confidence:** 4

**Summary:**

This paper studies the size generalization of GNNs in biological networks. Through the spectral analysis, the authors find that spectrum differences induced by size are related to differences in subgraph patterns (e.g., average cycle lengths). Since regular GNNs can hardly capture the cycle features, they propose three strategies, including self-supervision, augmentation, and size-insensitive attention, to enable GNNs to learn cycle information thus improving the OOD generalization across different sizes. Experiments with various GNN backbones show the proposed solutions can effectively improve their size OOD generalization ability.

**Strengths:**

(+) The spectral analysis along with the solutions are well-motivated and interesting to the community;

(+) The paper is well-written and easy to follow;

**Weaknesses:**

(-) The analysis especially the solutions lacks theoretical guarantees.

(-) The experiments focus on simple tasks and lack the comparison with several relevant baselines.

**Questions:**

1. The analysis especially the solutions lacks theoretical guarantees.
- Although the analysis shows that there is a connection between spectrum differences with the cycle lengths, there could be some underlying confounders that jointly affect the graph sizes and cycle lengths. For example, in the model by Bevilacqua et al. 2021, the graphon and the size of the graph will jointly affect the cycle lengths.
- The proposed three solutions are well motivated, while mainly based on empirical observations. To what extent can the three methods resolve the cycle issue? Will the operations affect the expressivity of GNNs?

2. The experiments focus on simple tasks and lack the comparison with several relevant baselines.
- Why do the experiments adopt a different data split scheme from previous practice such as in Bevilacqua et al. 2021?
- How well do the proposed methods perform on more realistic and large datasets such as OGB-molhiv with graph size shifts, and DrugOOD[1]?
- Can the proposed methods perform better than previous solutions like Bevilacqua et al. 2021, and Buffelli et al. that are cited in the paper, and [2,3] that are the state-of-the-art in graph size OOD generalizations?
- [4] analyzes the size generalization in link predictions, which is also a related work to discuss.
- Can the proposed methods improve the size generalization in algorithmic reasoning tasks?


**References**

[1] DrugOOD: Out-of-Distribution (OOD) Dataset Curator and Benchmark for AI-aided Drug Discovery -- A Focus on Affinity Prediction Problems with Noise Annotations, AAAI’23.

[2] Learning Causally Invariant Representations for Out-of-Distribution Generalization on Graphs, NeurIPS’22.

[3] Wasserstein Barycenter Matching for Graph Size Generalization of Message Passing Neural Networks, ICML’23.

[4] OOD Link Prediction Generalization Capabilities of Message-Passing GNNs in Larger Test Graphs, NeurIPS’22.

---

> ### Author Response · Authors · 2023-11-22
> **Response to Reviewer eDSS**
>
> We thank the reviewer for appreciating the novelty of the spectral analysis and the writing of this paper. We also thank the reviewer for carefully reading our paper and detailed comments. We appreciate all the questions and feedback the reviewer raised, especially on the number of related works provided. In the following, we carefully respond to each of the concerns and suggestions that the reviewer raised.
>
> **Q1.1 Other confounders that jointly affect the graph size and cycle lengths.**
>
> There might be other confounders that jointly affect the graph size and cycle lengths. However, the method by Bevilacqua et al. 2021 cannot fully address the covariate shifts we found. In that paper, they assume that the density of induced k-sized subgraphs is invariant. However, based on our empirical observations, we actually found big cycles that do not appear in small graphs, suggesting that this density invariance does not hold for large cycles. In other words, their model cannot generalize from cycles of small lengths to cycles of large lengths.
>
> **Q1.2 To what extent can the three methods resolve the cycle issue? Will the operations affect the expressivity of GNNs?**
>
> Explicitly including cycle information can help distinguish more graphs, but will not fundamentally affect the expressiveness of the model. To understand this problem better, in Table 4, we also include two expressive models as our baselines: RPGNN[R2] and SMP[R3]. What we find is that the general expressive model can help size generalization to some extent, but is not as good as explicitly providing this information. The model that excels at cycle-related tasks, e.g. SMP, exhibits better size generalizability than other expressive models, such as RPGNN.
>
> **Q1.3 Why do experiments adopt different data splits?**
>
> Firstly, it's important to note that we did not utilize identical datasets as those referenced in Bevilacqua et al. 2021. Secondly, the data files we obtained proved to be incompatible with the current versions of PyTorch and CUDA, preventing us from successfully loading the data into our environments. Thirdly, when attempting the data processing using the identical original code provided in Bevilacqua et al. 2021, we encountered errors. Consequently, we were unable to replicate the exact data split presented by Bevilacqua et al. 2021. Despite this deviation, our experimental setup closely aligns with the overarching concept of partitioning datasets into larger and smaller subsets.
>
> **Q1.4 Performance on other large-scale datasets**
>
> We sincerely thank the reviewer for giving several dataset examples. Initially, we explored several other biological datasets, such as the HIV dataset you mentioned. However, we encountered a significant class imbalance issue (please refer to the dataset section in the general response for details). For instance, in the HIV dataset, the 50% smallest dataset contains 20054 class 0 samples and 510 class 1 samples, whereas its 10% largest dataset contains 3680 class 0 samples and 433 class samples. Other large-scale datasets from ogb and TU exhibit similar trends. Additionally, for the other dataset mentioned in your review, DrugOOD includes various covariate shifts other than graph sizes. For the one concerning size shift (DrugOOD-lbap-core-ic50-size), it adopts a different idea in splitting the data, where it uses the large datasets in the training and validation set and small datasets in the testing. Due to such differences, we did not conduct experiments on it.
>
> **Q1.5 Comparison with baselines**
>
> We thank you for your great suggestions. We have added more baselines in Table 4. For more details, please refer to the baseline section in the general response and Table 4 in our updated submission. In general, the proposed method SIA still achieves the best performance and is model-agnostic.
>
> **Q1.6 Related work in link prediction OOD**
>
> Thank you for your great suggestion! We have added this paper to our related work section.
>
> **Q1.7 Size generalization in algorithmic tasks**
>
> One important view this paper takes is that the size-induced distribution shift depends on the data domain, and that is why contradictory conclusions appear in prior works. Though our main empirical findings apply mainly to biological data, we can adopt a similar spectral analysis to other fields, and identify other patterns that lead to the spectral differences.

---

> ### Comment · Reviewer_eDSS · 2023-11-22
>
> Thank you for the detailed explanation, which addressed many of my concerns. However, I still have some concerns:
> - What are the exact errors for loading the splits by Bevilacqua et al. 2021?
> - As acknowledged by the authors, the focus of the paper is on the biological data. The class imbalance issue generically exists in biological data, and would affect all methods. If the proposed method could indeed address the graph size shifts, then there are still improvements expected to be observed. Otherwise, without experiments on real data, the scope of the paper could be very limited;
> - I appreciate the authors' efforts in making the comparison with [3], but why [2] is neglected in comparison?

---

> > ### Author Response · Authors · 2023-11-23
> > **Response to Reviewer eDSS**
> >
> > We thank the reviewer sincerely for providing timely and constructive feedback.
> >
> >
> > **Q1 What are the exact errors for loading the splits by Bevilacqua et al. 2021?**
> >
> > The primary issue arises from the mismatch between our server's CUDA version and the specific PyTorch-Geometric (PyG) version required by the paper (Bevilacqua et al. 2021).
> >
> > This paper does not directly provide code files to generate the data splits. Instead, they provide .pt files to load data objects. Since they used an old PyG version, our current PyG cannot read these data objects, which raises the same errors as in these posts: https://github.com/pyg-team/pytorch_geometric/discussions/7241 and https://github.com/pyg-team/pytorch_geometric/issues/2040. We tried to install the old version of PyG, which involved the installation of old version of PyTorch, torch-scatter, and torch-sparse. However, pip-install these old packages from the source did not solve this issue. The reason is that the dependency of PyG (torch-scatter/torch-sparse) requires the old CUDA. We also tried the solution suggested by these posts to export LD_LIBRARY_PATH from the installation of the old PyTorch, which did not work on our end. We’ve consulted PyG engineers for alternative solutions. The best solution they offered us is to install a supportive old CUDA toolkit from the source. However, this also failed, because the required CUDA version is too old to be supported on our server system.
> >
> > As a consequence, we simply follow the same data split idea and follow closely as what has been written in Bevilacqua et al. 2021.
> >
> > **Q2 Comparison with CIGA([2])**
> >
> > Thanks for your suggestion. We have just included the performances of two versions of CIGA (Learning Causally Invariant Representations for Out-of-Distribution Generalization on Graphs, NeurIPS’22) in Table 4 in our second update of the paper. CIGA is indeed a strong baseline, but we can still see that the average improvements across different datasets are not as good as SIA.
> >
> > **Q3 Performance on real-world data**
> >
> > We have also tested our method on the hiv dataset and provided the results in Appendix H. There are still clear performance gains after applying our methods. Due to limited time to run large datasets, we were not able to run other baseline strategies.

---

> ### Comment · Reviewer_eDSS · 2023-11-23
>
> Thank you for the new results and discussion! I have increased my rating accordingly. I believe this work could be more impactful if the developed insights and methods could be generalized to realistic datasets.

---

> > ### Author Response · Authors · 2023-11-23
> >
> > Thank you for your valuable suggestions and quick responses!

---

### Author Response · Authors · 2023-11-22
**General Response (Part I)**

We express our gratitude to all reviewers for their insightful feedback and valuable suggestions, which have significantly contributed to the improvements of our paper. We will begin by answering general questions and then address the specific questions posed by individual reviewers. The paper has been updated and important changes are marked red. **We kindly request the reviewers to download the updated version of our submission (including the main paper and the supplementary file).**


**Scope and motivation of this paper.** Our primary goal is to offer empirical insights into the true distribution shifts resulting from variations in size. It's worth to note that our intention is not to introduce a novel methodology for the general Out-of-Distribution (OOD) problem. These empirical insights are important as contradictory conclusions [1-2] exist stemming from different assumptions and application domains. Therefore, our emphasis is on the biological domain, which encompasses the majority of datasets for graph classification. Our main contribution is that we find cycle-related information (existence of cycles, average cycle length) is critical for size generalization for biological data, which can inspire future GNN designs. We substantiate this insight by integrating such information into several commonly employed designs, all of which consistently showcase improvements. Thanks for reviewer dD3c for bringing an interesting view that expressive models that excel at capturing cycle information may exhibit good size generalizability. We added more baselines in Table 4 and found that this indeed is true, which further validates our empirical findings.

[1] Transferability of spectral graph convolutional neural networks. JMLR'21

[2] From local structures to size generalization in graph neural networks. ICML'21

**Baselines.** We appreciate the baselines provided by the reviewer eDSS and dD3c. We have added the following baselines in Table 4 (marked red):

[r1] "sizeshiftreg: a regularization method for improving size-generalization in graph neural networks", NeuralPS 2022

[r2] "Relational pooling for graph representations", ICML 2019.

[r3] "Building powerful and equivariant graph neural networks with structural message-passing" NeurIPS 2020.

**[Update on Nov. 22]**
[r4] Learning Causally Invariant Representations for Out-of-Distribution Generalization on Graphs, NeurIPS’22.

[r1], [r2] and [r4] focus directly on size Out-of-Distribution (OOD), while [r2] and [r3] propose more expressive models. Additionally, [r1], [r2] and [r4] are model-agnostic, and [r3] excels at detecting cycles. Analyzing the results presented in Table 4, we have the following conclusions: (1) the performances of [r1-r4] consistently lag behind ours; (2) the subgraph-aware causual model [r4] is the second best strategy; and (3) [r3], with its advantage in detecting cycles, achieves notable performance, particularly evident in the bbbp dataset. This observation aligns with the empirical insights from our paper, highlighting the significance of cycle information in size generalization. Though [r3] excels on certain dataset, it is worth mentioning that directly providing cycle information (e.g., our proposed strategy SIA) is generally better. Expressive models do not universally excel at capturing cycle information, elucidating why [r2] does not exhibit impressive performance.

For other baselines brought by the reviewers that we did not compare with, [r5-r6] do not provide their codes, [r7] misses important code files, and [r8-r10] are not directly related to the scope of this paper.

- [r5] Wasserstein Barycenter Matching for Graph Size Generalization of Message Passing Neural Networks, ICML'23
- [r6] From local structures to size generalization in graph neural networks, ICML'21
- [r7] Size-invariant graph representations for graph classification extrapolations, ICML'21
- [r8] From stars to subgraphs: Uplifting any GNN with local structure awareness. ICLR'22
- [r9] "Improving graph neural network expressivity via subgraph isomorphism counting", IEEE TPAMI'22
- [r10] DrugOOD: Out-of-Distribution (OOD) Dataset Curator and Benchmark for AI-aided Drug Discovery -- A Focus on Affinity Prediction Problems with Noise Annotations, AAAI’23.

---

### Author Response · Authors · 2023-11-22
**General Response (Part II)**

**Datasets.** The datasets utilized in this paper do not demonstrate significant class imbalance, with the major class being no more than ten times larger than the minor class. While we appreciate some suggestions from other reviewers, many large-scale datasets from Tu-datasets and OGB suffer from significant issues of class imbalance. We present the statistics below for some of the commonly used large-scale datasets, including the one from reviewer eDSS:

| Dataset | Number of Small 50% class 0 samples | Number of Small 50% class 1 samples | Number of Large 10% class 0 samples | Number of Large 10% class 1 samples |
| ------- | --------------------------- | --------------------------- | --------------------------- | --------------------------- |
| Hiv     | 20054                       | 510                         | 3680                        | 433                         |
| MOLT-4  | 18860                       | 1023                        | 3284                        | 69                          |
| MCF-7   | 13138                       | 747                         | 2243                        | 534                         |
| Yeast   | 36258                       | 3543                        | 6553                        | 1408                        |
| PC-3    | 13316                       | 439                         | 2356                        | 395                         |

This significant class imbalance leads to two problems. First, the backbone GNNs (e.g. GCN, GAT) generalize poorly even on in-distribution data. It does not make sense to study size OOD in those scenarios. Second, this imbalance usually leads to a big difference in label distribution between training and test data. This can mislead the results if the model is biased toward predicting certain class which has a larger portion in the test set.


**Theoretical guarantee.** Our findings are rooted in empirical observations, and our insights are solely derived from data. Modeling how cycle-related information changes with size may necessitate domain knowledge, which, unfortunately, falls beyond the scope of this paper. The theoretical insights for our investigation into cycles is based on our discernment of variations in the peaks of specific frequencies within the graph spectrum, such as the frequency=1 in unnormalized Laplacian matrices. Many of these specific frequencies align with the frequencies of cycles, with cycles containing 5-7 nodes being particularly evident. As a reminder, the spectrum for a cycle with n nodes is given by the formula 2-2cos(2πj/n).

---

### Meta-Review · Area_Chair_DA3g · 2023-12-05

**Metareview:**

This paper investigates size generalization in Graph Neural Networks (GNNs) within biological networks. Through spectral analysis, it identifies a link between size-related spectrum differences and subgraph patterns, like average cycle lengths. To address the challenge that standard GNNs face in capturing cycle features, the study proposes three strategies: self-supervision, augmentation, and size-insensitive attention. These methods aim to improve GNNs' ability to learn cycle information, thereby enhancing their out-of-distribution (OOD) generalization across various network sizes. Experimental results with different GNN backbones validate the effectiveness of these strategies in enhancing size generalizability in GNNs, particularly for biological data.

**Justification For Why Not Higher Score:**

The main criticisms of the paper revolve around its conceptual novelty and relation to prior work. A significant question raised is the impact of cycle distribution changes on vanilla Graph Neural Networks (GNNs), given that these GNNs are inherently limited in recognizing cycles. The critics argue that if vanilla GNNs cannot detect cycles, then the changes in cycle distribution will not directly affect their performance.

Furthermore, the study is seen as a subcase of previous research "From local structures to size generalization in graph neural networks" (referred to as [1]), which did not just consider degree patterns but also d-patterns that directly correspond to the information GNNs process during message passing. This implies that any change in connectivity induced by graph size alterations should be visible within these d-patterns. Therefore, the effect observed in this paper could either be a specific instance of [1] (if the changes in cycle distribution are deducible from d-patterns) or the paper's claim might be incorrect (if the stated differences do not impact GNN computations).

This critique suggests the possibility of other confounding factors that concurrently influence graph size and cycle lengths, indicating that the paper's findings might not be as straightforward as presented. The critics imply that a deeper analysis might be required to understand the true nature of the observed phenomena and their implications for GNNs.

**Justification For Why Not Lower Score:**

N/A

---

### Decision · Program_Chairs · 2024-01-16

Reject